# Cadmium binding by the F-box domain induces p97-mediated SCF complex disassembly to activate stress response programs

Linda Lauinger[1] ✉, Anna Andronicos[1], Karin Flick[1], Clinton Yu[2], Geetha Durairaj[1], Lan Huang [2] & Peter Kaiser [1] ✉

The F-box domain is a highly conserved structural motif that defines the largest class of ubiquitin ligases, Skp1/Cullin1/F-box protein (SCF) complexes. The only known function of the F-box motif is to form the protein interaction surface with Skp1. Here we show that the F-box domain can function as an environmental sensor. We demonstrate that the F-box domain of Met30 is a cadmium sensor that blocks the activity of the SCF$^{Met30}$ ubiquitin ligase during cadmium stress. Several highly conserved cysteine residues within the Met30 F-box contribute to binding of cadmium with a $K_D$ of 8 μM. Binding induces a conformational change that allows for Met30 autoubiquitylation, which in turn leads to recruitment of the segregase Cdc48/p97/VCP followed by active SCF$^{Met30}$ disassembly. The resulting inactivation of SCF$^{Met30}$ protects cells from cadmium stress. Our results show that F-box domains participate in regulation of SCF ligases beyond formation of the Skp1 binding interface.

Cells must be able to accurately assess their environment for the availability of essential nutrients and potential threats. Thus, systems that detect the abundance of harmful substances are fundamental to protect the organism. The ubiquitin-proteasome system (UPS) is well known as a dynamic regulator of protein abundance as well as non-proteolytic signaling[1–5]. Therefore, it is not surprising that the UPS plays important roles in numerous biological pathways, including various cellular stress response programs[6]. Ubiquitin conjugation requires the E1-E2-E3 enzyme cascade in which the final step of substrate-specific ubiquitylation is carried out by the most diverse components in the system, the E3 ubiquitin ligases[1]. E3s can be classified in different subgroups of which the multi-subunit Cullin-RING complexes (CRLs) are the largest representative[7,8]. The CRL subfamily of SCF (Skp1-Cullin-F-box) ligases are composed of the RING finger protein Rbx1, the scaffold Cul1 (yeast Cdc53), and Skp1, which connects an array of substrate-specific F-box proteins to the core ligase[9–11]. The defining structural feature of SCF ubiquitin ligases is a short, highly conserved motif known as the F-box domain, which mediates the Skp1/F-box interaction[12,13].

Typically, SCF ligase activity is regulated at the level of substrate binding[9,14]. However, signal-induced, subtype-selective SCF complex disassembly and modulation of the F-box/Skp1 interface by kinases has also been demonstrated as a novel mechanism of SCF regulation[15–20]. This mechanism is best characterized in the yeast SCF$^{Met30}$ ubiquitin ligase. The two critical substrates of SCF$^{Met30}$ are the transcriptional activator Met4 and the cell cycle inhibitor Met32[15,21–23]. Together they coordinate the initiation of a transcriptional response during nutritional or cadmium stress, which ensures cellular integrity, restoration of sulfur-containing metabolites, and the synthesis of glutathione for protection against cadmium[17,18,21,24–27]. Under normal growth conditions, both Met4 and Met32 are ubiquitylated by SCF$^{Met30}$ with a lysine-48-linked chain. Even though this presents a classical destruction signal, only Met32 is directed to the 26S Proteasome. Met4 is kept in a stable but transcriptionally inactive state by ubiquitylation[22,28,29].

[1]Department of Biological Chemistry, University of California, Irvine, Irvine, CA 92697, USA. [2]Department of Physiology and Biophysics, University of California, Irvine, Irvine, CA 92697, USA. ✉e-mail: llauinge@uci.edu; pkaiser@uci.edu

SCF^Met30-dependent ubiquitylation of Met4 and Met32 are blocked during nutritional as well as cadmium stress. The mechanisms to control ligase activity are however profoundly different (Fig. 1A). During nutrient stress, the interaction between Met4 and SCF^Met30 is prevented and substrate ubiquitylation cannot occur. Intriguingly, cadmium specifically leads to the dissociation of the F-box protein Met30 from Skp1, and thus the core ligase (cullin-Skp1), by triggering autoubiquitylation of Met30[17,18]. This ubiquitin chain serves as a recruitment signal for the AAA-ATPase Cdc48/p97 and its cofactor Shp1, which actively extract Met30 protein from the cullin-Skp1 complex[15,30,31]. Dissociated "Skp1-free" Met30 is then degraded by the non-canonical Cdc53/Rbx1 ligase via the proteasomal pathway[32].

SCF^Met30 is thus a hub that coordinates sulfur amino acid home-ostasis and the cellular response to cadmium stress. How SCF^Met30 senses cadmium or sulfur-containing metabolism is unknown. Here we

show that the cadmium sensing element in SCF^Met30 is the F-box domain itself. Multiple key cysteine residues in the F-box motif of Met30 enable selective cadmium binding, which triggers a cascade of downstream events to protect the organism during cadmium stress. Our results show that F-box domains can play a direct role in per-ceiving environmental cues and connecting them to cellular response pathways.

## Results

### Highly conserved cysteine-rich domains in Met30 as candidate cadmium sensors

The yeast ubiquitin ligase SCF^Met30 coordinates a dedicated signaling pathway in response to the status of sulfur amino acid metabolism and cadmium stress (Fig. 1A). How SCF^Met30 detects metabolites or cad-mium is unknown. In order to define mechanisms of possible direct

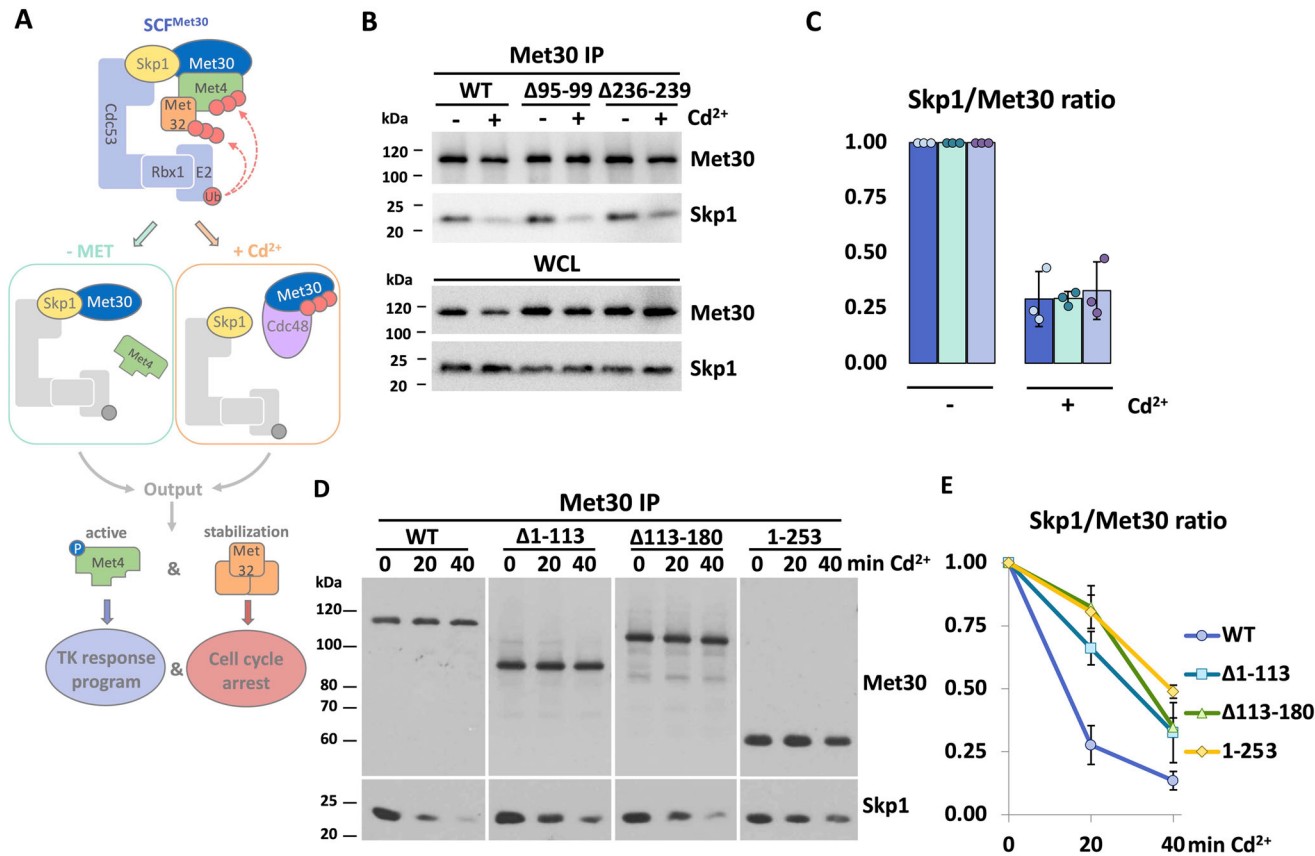

**Fig. 1 | Highly conserved cysteine rich domains in Met30 as candidate cadmium sensors. A** The E3 ligase SCF^Met30 orchestrates the response programs during either nutritional or heavy metal stress, however through profound distinct mechanisms. Nutritional stress: Under normal growth conditions, high levels of methionine and metabolites promote a stable complex between the transcriptional activator Met4 and the E3 ligase via the F-box protein Met30. Ubiquitinylated Met4 is stable and transcriptionally inactive. Low methionine or SAM levels block binding of the substrate Met4 to SCF^Met30. Heavy metal stress: Under normal growth conditions, the F-box protein Met30 forms a stable complex with Skp1, facilitating the ubi-quitylation of its substrates. Cadmium exposure triggers the autoubiquitylation of the F-box protein Met30, which is a recruitment signal for the segregase Cdc48. The active dissociation of Met30 from Skp1 by Cdc48 results in the inactivation of the ligase. Both stress conditions result in a shared response program. Met4 is deubi-quitylated and phosphorylated and becomes transcriptionally active. A transcrip-tional response program to restore sulfur-containing amino acids or components for detoxification is initiated. Additionally, the accumulation of the cell cycle inhibitor Met32 leads to a cell cycle arrest during these stress conditions. **B** The deletion of either proposed metal binding motif does not impair cadmium induced disassembly of SCF^Met30. Cells expressing ^12xMycMet30 WT,

Δ95–99, or Δ236–239 were cultured at 30 °C in YEPD medium and treated with 100 μM CdCl₂ and samples were harvested after 30 min of exposure. Native whole cells lysates (WCL) were prepared and ^12xMycMet30 was immunoprecipitated (Met30 IP) and co-precipitated Skp1 was analyzed by western blot. **C** Densitometric analysis of western blot band intensities of immunoprecipitations. For quantifications the signal intensities for each Met30 variant at time point 0 were set to 1 and Skp1 signals were normalized to ^12xMycMet30 to quantify Met30 dissociation from the core ligase (n = 3 independent experiments), data are represented as mean ± SD. **D** Strains were cultured at 30 °C in SDC-LEU medium and treated with 100 μM CdCl2, and samples were harvested at indicated time points. Native whole cell lysates (WCL) were prepared and ^12xMycMet30 was immunoprecipitated (Met30 IP) and co-precipitated Skp1 was analyzed by western blot. **E** Densitometric analysis of western blot band intensities of immunoprecipitations. For quantifications, the signal intensities for each Met30 variant at time point 0 were set to 1 and Skp1 signals were normalized to ^12xMycMet30 to quantify Met30 dissociation from the core ligase (n = 3 independent experiments), data are represented as mean ± SD. Results shown in **B**, **C**, **D** are representative blots from three inde-pendent experiments. Source data are provided as source data files for figures C and E.

cadmium binding we focused on evolutionarily conserved cysteine-containing domains, as cysteine residues usually facilitate protein/metal interactions[33]. We identified two highly conserved cysteine-containing motifs (residues 95-99 and 236–239) that were good candidates for cadmium sensors (Supplementary Fig. 1A). Residues 95-99 proximal of the dimerization domain and residues 236–239 distal of the F-box motif were deleted individually at the genomic locus using a CRISPR/Cas9 approach, and cells were exposed to either methionine starvation or cadmium stress. Under normal growth conditions SCF[Met30] ubiquitylates the transcriptional activator Met4 and therewith represses both the cadmium and methionine-starvation-related stress responses. Cadmium stress triggers the dissociation of Met30 from the Cul1-Skp1 complex, whereas low methionine levels lead to the separation of Me4 from SCF[Met30] (Fig. 1A). Both events result in deubiquitylated and active Met4, leading to cell cycle arrest and the induction of the respective transcriptional response programs. Surprisingly, deletion of either domain completely blocked the response to methionine stress, while only modestly delaying the cadmium stress response (Supplementary Fig 1B). These results suggested that the cysteine rich conserved regions 95-99 and 236–239 are important for sensing the status of sulfur amino acid metabolism, but not cadmium stress. To confirm the latter, we evaluated cadmium-induced SCF[Met30] disassembly in the Δ95–99 and Δ236–239 mutants. Met30 dissociation from the Cul1-Skp1 complex was unaffected by deletion of residues 95–99 or 236–239 (Fig. 1 B, C). To our surprise, these results exclude regions 95–99 and 236-239 as cadmium sensors, but instead suggest their role in sensing metabolic cues.

The above experiments excluded these conserved cysteine-rich regions as cadmium sensors, and we therefore initiated an unbiased approach by analyzing various truncation mutants of Met30. Neither deletions of the N-terminus (Δ1–113), the dimerization domain (Δ113-180), or the WD40-containing C-terminus (only aa 1-253 were expressed; Met30[1-253]) blocked cadmium sensing as measured by dissociation of Met30 from Skp1 (Fig. 1 D, E, Supplementary Fig. 1C, D). Some truncation mutations showed delayed dissociation, but cadmium was still detected. We reasoned that the slower kinetics could be due to involvement of the relatively large deletions with steps downstream of cadmium sensing, such as Met30 autoubiquitylation, or recruitment of Cdc48 and/or recruitment of Shp1[15,30].

## Highly conserved cysteine residues in the F-box are unique to Met30 family members

The truncation experiments likely excluded all but the F-box domain as the cadmium sensor in Met30. Intriguingly, an alignment of all characterized F-Box proteins in yeast revealed that Met30 contains the only F-box with high cysteine abundance (Fig. 2A). Furthermore, these cysteine residues are highly conserved throughout Met30 orthologs (Fig. 2B). All these observations suggested that the F-box domain in Met30 could function as a cadmium sensor. However, we could not employ the same strategy of analyzing Met30/Skp1 dissociation to define cadmium sensing capabilities of the F-box region, as the F-box is absolutely essential for the Met30/Skp1 interaction. We therefore exchanged the F-box motif of Met30 with those from the related F-box proteins Cdc4, Grr1, and Mfb1 and tested cadmium-induced dissociation of these hybrid Met30 proteins from Skp1. While the wild-type Met30 efficiently dissociated from Skp1 during cadmium stress, swapping F-box domains from either Cdc4, Grr1, or Mfb1 completely blocked dissociation (Fig. 2C). The non-native F-box domains from Cdc4 and Mfb1 mediated generally weaker binding to Skp1 than the native Met30 F-box, and the Grr1 F-box decreased overall Met30[Grr1] stability, supporting that F-box domains are not simply interchangeable and evolved in the context of their surrounding sequences. Although the results obtained with these swapped in F-box domains are consistent with the idea of the Met30 F-box functioning as a cadmium sensor, the effects of non-native F-box domains on overall

Met30 function complicated interpretation of results (Fig. 2C). Met30[Mfb1] displayed the best compromise of protein stability and complex integrity (Supplementary Fig. 2). We therefore monitored disassembly kinetics in this hybrid mutant (Fig. 2D). This experiment further supported the model that the Met30 F-box is required to detect cadmium, because even after 60 min of cadmium exposure, SCF[Met30] disassembly was not induced.

## Cadmium sensing by the Met30 F-box domain is important to mediate the cadmium stress response

Although the F-box domain swap experiment is suggestive, to obtain more direct evidence for the role of the Met30 F-box domain in cadmium sensing we mutated the three conserved cysteine residues to serine by CRISPR/Cas9 mediated genome editing. However, when cysteine residues 201, 205, and 211 were simultaneously mutated to serine, the steady state interaction of Met30 with Skp1 was significantly decreased (Supplementary Fig. 3A). Notably, exposure to cadmium did not lead to further dissociation of Met30 from Skp1, but the compromised interaction of the serine mutants made it difficult to draw unambiguous conclusions. We therefore generated individual cysteine mutations. Consistent with these three F-box cysteines participating in cadmium sensing, the individual cysteine mutants were less effective in responding to cadmium stress because they maintained a significant fraction of Met4 in the ubiquitylated, repressed state after cadmium exposure (Fig. 3A). Furthermore, changing any of the three cysteine residues blocked cadmium-induced dissociation of Met30 from Skp1 in vivo, providing a rationale for continued Met4 ubiquitylation (Fig. 3B and Supplementary Fig. 3B). The suppression of cadmium-induced SCF[Met30] disassembly by the individual cysteine mutations predicts that the Met4-controlled transcriptional stress response program is disrupted. Indeed, induction of the Met4-dependent genes *MET25* and *GSH1* was blunted (Fig. 3C). Importantly, expression of these genes was unaffected in response to metabolic stress signals indicating that all aspects of regulating the SCF[Met30]/Met4 system, except cadmium sensing, are fully functional (Fig. 3D).

Defects of SCF[Met30] to detect or respond to cadmium stress result in blocked induction of a defense program mediated by Met4-dependent gene transcription. Most notably, induction of *GSH1*, which encodes the rate-limiting enzyme for glutathione synthesis, depends on Met4[15,17,18]. Yeast cells lacking this response mechanism are hypersensitive to cadmium stress. Neither of the cysteine mutations resulted in a growth defect under normal growth conditions indicating that they are not required under non-stress conditions (Fig. 3E). However, mutation of these F-box cysteines rendered cells sensitive to cadmium exposure, further supporting their importance for the cadmium stress response (Fig. 3E and Supplementary Fig. 3C).

Our previous studies have demonstrated that when Cdc48/p97 activity is blocked during cadmium stress, Met30 is autoubiquitylated, but remains bound to Skp1, prone for extraction by the segregase[15]. Thus, mechanistically cadmium does not directly disrupt the Met30/Skp1 interaction, but rather induces Met30 autoubiquitylation, which in turn recruits Cdc48/p97 and its cofactor Shp1. Together they actively dissociate Met30 from Skp1[15,30]. Disassembly of SCF[Met30] was attenuated in all cysteine mutants, and we therefore analyzed upstream events that lead to Met30 extraction from the core SCF complex. All mutants that blocked dissociation showed decreased levels of Cdc48 recruitment during cadmium stress (Fig. 4A and Supplementary Fig. 3D, E). Segregase activity is required for complex dissociation and reduced Cdc48 recruitment can thus explain diminished Met30 dissociation. We next asked why Cdc48 recruitment was impaired in Met30 mutants during cadmium stress. Met30 autoubiquitylation is the recruitment signal for Cdc48[15]. All mutants show significantly decreased levels of autoubiquitylation compared to wild type (Fig. 4B and Supplementary Fig. 3F–H). Thus, lower levels of

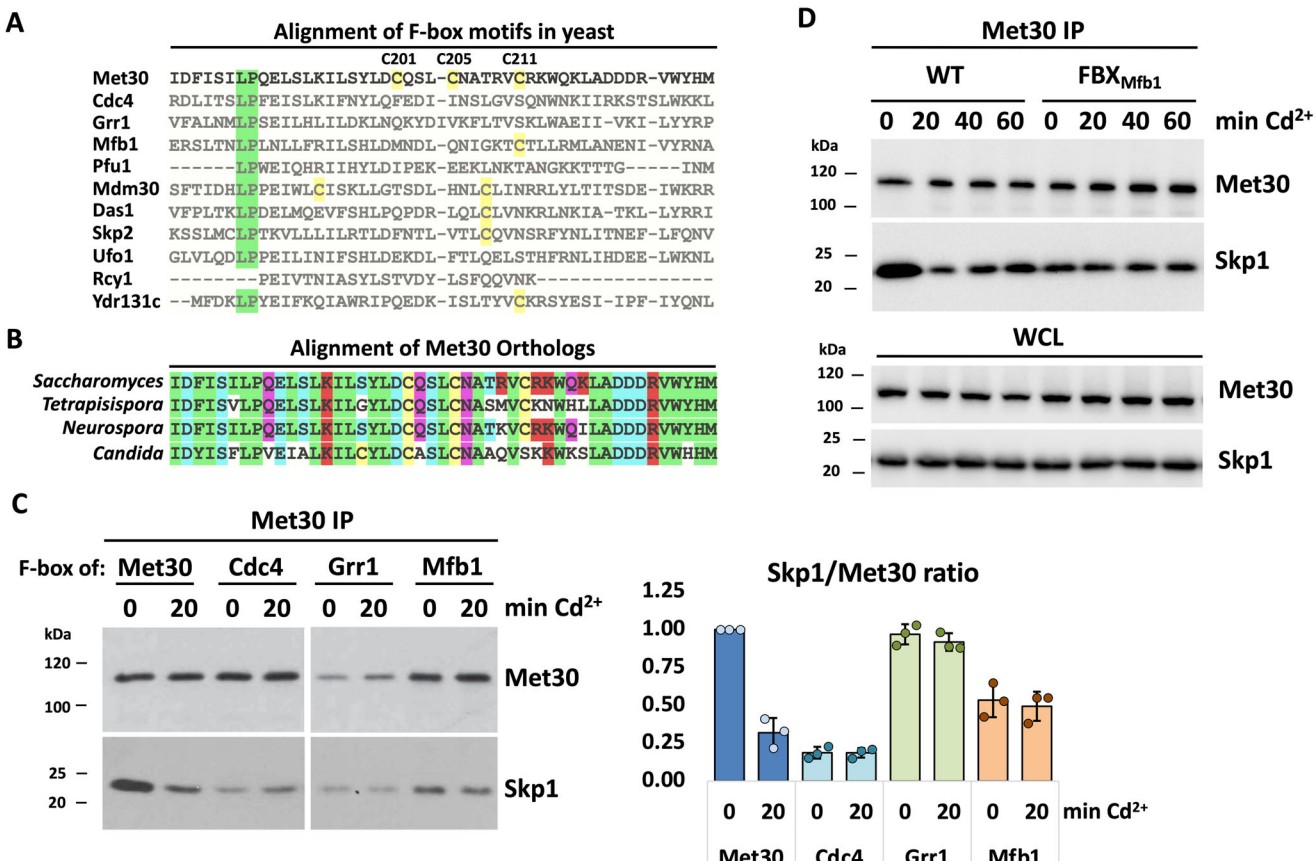

**Fig. 2 | Highly conserved cysteine residues in the F-box of Met30. A** Met30 is the only F-box protein in yeast that displays high abundance of cysteine residues in the F-box motif. Alignment of the F-box motif of characterized F-box proteins in *Saccharomyces cerevisiae*. Highly conserved amino acids in green, and cysteine residues highlighted in yellow. **B** The cysteine residues in the F-Box of Met30 are highly conserved throughout fungi. Alignment of Met30 orthologs *Saccharomyces, Candida, Tetraspora* and *Neurospora*. Cysteine residues highlighted in yellow. **C** Replacing the F-box motif of Met30 prevents cadmium-induced dissociation of SCF^Met30. Exchange of Met30's F-box motif blocks cadmium-induced dissociation from the core ligase. Cells expressing ^{12xMyc}Met30 WT or Met30 with the F-box motif of Cdc4, Grr1, and Mfb1, respectively, were cultured at 30 °C and treated with

100 μM CdCl$_2$ for 20 min. Native whole cells lysates were prepared and ^{12xMyc}Met30 was immunoprecipitated (Met30 IP) and co-precipitated Skp1 was analyzed by western blot. For quantification, the signal intensity ratio of Skp1/Met30 WT at timepoint 0 was set to 1 and Skp1 signals were normalized to ^{12xMyc}Met30 to quantify Met30 dissociation from the core ligase (*n* = 3 independent experiments), data are represented as mean ± SD. **D** Longer cadmium exposure does not trigger dissociation of Met30 FBX$_{Mfb1}$ from the core ligase. Experiment was performed as above but for extended time. Results shown in **C**, **D** are representative blots from three independent experiments. Source data are provided as source data files for (**C**).

Cdc48 enrichment in the complex are likely caused by the autoubiquitylation defects in all mutants.

The role of cadmium is thus the induction of Met30 autoubiquitylation to initiate SCF^Met30 disassembly. We hypothesized that cadmium induces a conformational change in Met30 to reposition it for autoubiquitylation. To probe for a potential cadmium-induced conformational change we analyzed SCF^Met30 before and after cadmium exposure by cross-linking mass spectrometry. Cross-linking was achieved with disuccinimidyl sulfoxide (DSSO), a MS-cleavable, amine targeting chemical crosslinker, with two symmetric collision-induced dissociation-cleavable sites that allowed more effective identification of cross-linked peptides by mass spectrometry[34]. Consistent with our model, we observed a striking change in the spectrum of cross-linked peptides upon cadmium exposure, indicative of a conformational change (Fig. 4C, Supplementary Fig. 3I and Supplementary Data 1).

**Cysteine 228 proximal to the F-box provides the fourth cadmium ligand**
Most cadmium binding sites require four ligands[35]. Within the F-box motif there are no further candidates for a potential fourth binding site. However distal of the motif we found two highly conserved residues (H226 and C228) that might function as a fourth ligand. We used

CRISPR/Cas9 editing to generate H226R and C228S point mutations. Since the H226R mutation was lethal in a wild-type background, a met32 knockout was used to prevent Met30-induced cell cycle arrest via Met32 stabilization[36]. The H226R mutant responded with dissociation of Met30 to cadmium stress excluding a role of H226 in cadmium sensing (Supplementary Fig. 4A). In contrast, in C228S mutant cells Met4 remained ubiquitylated and dissociation of Met30 from the core ligase was significantly attenuated (Fig. 4D, E, Supplementary Fig. 4B). Furthermore, induction of the Met4-dependent gene *MET25* was significantly reduced upon cadmium exposure. Expression in response to methionine starvation was unaffected confirming the overall integrity of SCF^Met30/Met4 activity (Fig. 4F). In addition, C228S cells exhibit cadmium hypersensitivity, further supporting a role for cysteine 228 as the fourth cadmium binding ligand (Supplementary Fig. 4C, D).

We next tested whether the cadmium-sensing cysteines may be prone to oxidation and oxidative stress can interfere with cadmium sensing. Note that exposing cells to oxidative stress does not trigger a response of the SCF^Met30/Met4 system[17]. Cells were pre-exposed to H$_2$O$_2$ for 40 min to trigger oxidative stress followed by cadmium exposure. Both, cadmium-induced dissociation of Met30 and the downstream transcriptional response program were blocked by H$_2$O$_2$

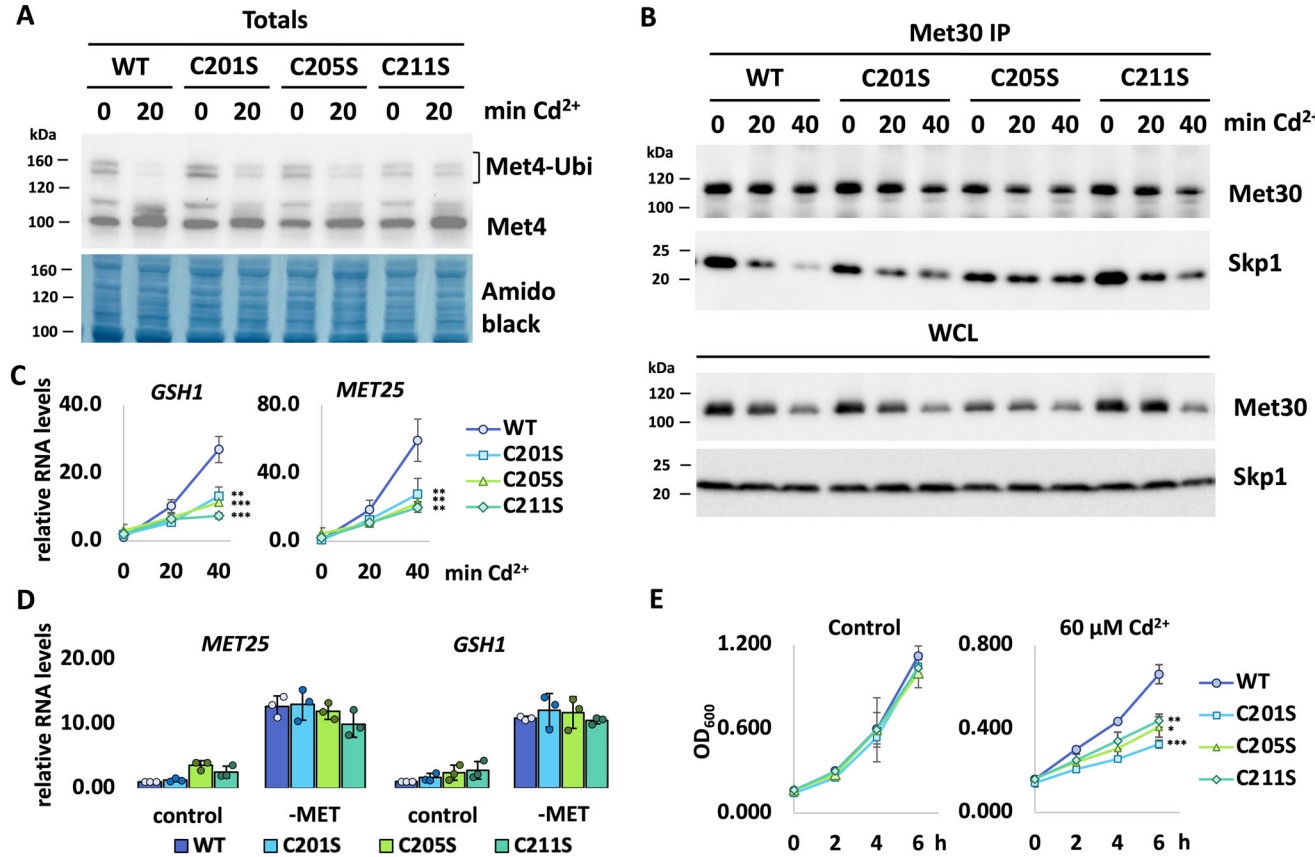

**Fig. 3 | Single cysteine mutations in F-box motif attenuate cadmium-induced dissociation and downstream effects. A** Met4 deubiquitylation in single cysteine mutants is blocked. Strains were cultured in YEPD media, treated with 100 μM CdCl₂, and samples were harvested after 20 min of exposure. Whole-cell lysates were analyzed by western blot using a Met4 antibody to follow the ubiquitylation status of Met4. **B** Dissociation kinetics of Met30 from the SCF core ligase are decreased in single cysteine mutants. Cells were cultured at 30 °C in YEPD medium and treated with 100 μM CdCl₂ and samples were harvested at indicated time points. ¹²ˣᴹʸᶜMet30 was immunoprecipitated (Met30 IP) and co-precipitated Skp1 was analyzed by western blot. **C** Decreased cadmium-induced gene expression in single cysteine F-box mutants. RNA was prepared from samples shown in (**B**) and expression of Met4 target genes *MET25* and *GSH1* was analyzed by RT-qPCR and normalized to 18S rRNA levels (*n* = 3 independent experiments), data are represented as mean ± SD, *p* values calculated by two-tailed student's *t*-test: *$p$ < 0.1, **$p$ < 0.05, ***$p$ < 0.01. *GSH1*: C201S = 0.010, C205S = 0.024, C211S = 0.010. *Met25*: C201S = 0.012, C205S = 0.031, C211S = 0.006. **D** The expression of Met4-dependent

genes in response to methionine starvation is unaffected in single cysteine mutants. Depicted yeast strains were grown at 30 °C in YEDP media. For methionine starvation, cultures were washed with water and shifted to minimal medium without methionine for 30 min and then harvested. RNA was extracted and expression of Met4 target genes *MET25* and *GSH1* was analyzed by RT-qPCR and normalized to 18S rRNA levels (*n* = 3 independent experiments), data are represented as mean ± SD. **E** Single Cysteine mutations in F-box mutants indicate cadmium sensitivity. Cells expressing endogenous ¹²ˣᴹʸᶜMet30 WT or C201S, C205S, C211S respectively were cultured in YEPD medium in the absence and presence of 60 μM CdCl₂ and samples were taken at indicated time points to measure optic density at 600 nm (*n* = 3 independent experiments), data are represented as mean ± SD, *p* values calculated by two-tailed student's t-test: *$p$ < 0.1, **$p$ < 0.05, ***$p$ < 0.01. 6 h timepoint: C201S = 0.009, C205S = 0.060, C211S = 0.013. Results shown in **A**, **B** are representative blots from three independent experiments. Source data are provided as source data files for (**C**, **D**, **E**).

priming (Supplementary Fig. 4E–G). We considered that H₂O₂ might induce glutathione production and prepare cells for cadmium stress. However, H₂O₂ concentrations used did not induce glutathione production as demonstrated by unchanged expression levels of the rate-limiting enzyme in the glutathione pathway, Gsh1 (Supplementary Fig. 4F). In addition, *Δgsh1* deletion cells remained unresponsive to cadmium stress when pre-exposed to H₂O₂ (Supplementary Fig. 4F, G). These results suggest that oxidative stress may change the oxidation state of one or more cadmium-sensing cysteines in Met30 and thereby block cadmium sensing. However, we cannot exclude that H₂O₂ priming induces antioxidant responses other than glutathione and decreases free cadmium levels.

**The F-box motif in Met30 mediates specific binding to cadmium in vitro**
To obtain more direct evidence for the role of the Met30 F-box domain in cadmium sensing, we measured the cadmium binding capability of Met30. To this end, we generated a Cd-NTA resin (Fig. 5A) and

performed affinity chromatography with whole cell lysates (WCL) denatured in 8 M urea (Fig. 5). As demonstrated by Immobilized Metal Ion Affinity Chromatography (IMAC), metal binding protein domains can maintain binding specificity under fully denaturing conditions. We therefore initially used denaturing conditions to avoid indirect binding to Cd-NTA through Met30 binding partners. Met30 readily purified on Cd-NTA (Supplementary Fig. 5A), and binding was specific as the F-box protein Mfb1 did not bind to Cd-NTA (Fig. 5B). Consistent with intact cadmium sensing of truncation mutants that lack the N-terminus (Δ1–113), the dimerization domain (Δ113–180), or the WD40-containing C-terminus (Met30¹⁻²⁵³), these mutants bound to Cd-NTA with similar affinity as wild-type Met30 (Fig. 5C). The only common region in all these Met30 truncation mutants is the F-box domain further supporting location of the cadmium binding region in the F-box motif. We next compared Met30 binding to Cd-NTA in the absence and presence of the F-box motif. These experiments were done in the context of Met30 lacking the WD40 domain (Met30¹⁻²⁵³). This Met30 fragment showed full binding to cadmium (Fig. 5D).

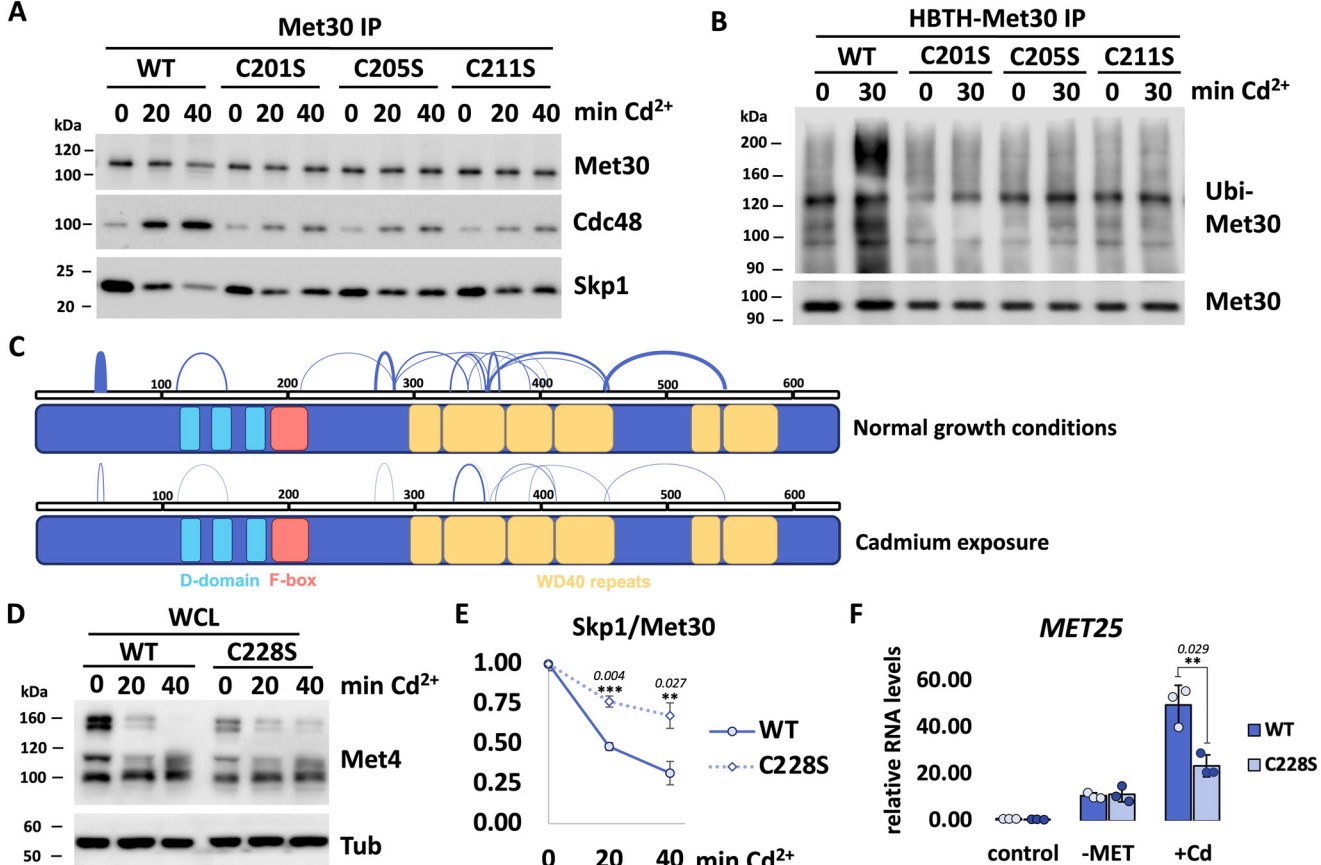

**Fig. 4 | Cadmium induces Met30 autoubiquitylation to initiate SCF^Met30 disassembly. A** Cdc48 recruitment in single cysteine mutants is reduced. ^12xMycMet30 was immunoprecipitated (Met30 IP) and co-precipitated ^RGS6HCdc48 and Skp1 was analyzed by western blot. **B** Cadmium-induced autoubiquitylation of Met30 is decreased in single cysteine mutants. Cells expressing ^HBTHMet30 were cultured at 30 °C in YEPD medium and treated with 100 μM CdCl$_2$ and samples were harvested after 30 min. Met30 variants were purified on Ni$^{2+}$-NTA Sepharose under denaturing conditions and analyzed by western blot using antibodies directed to ubiquitin or streptavidin-HRP to detect HBTH-Met30. Cells in which *met30* and *met4* are knocked out were used as a control to define unspecific ubiquitin background precipitation (Supplementary Fig. 3G). **C** Cadmium binding to the F-box likely induces a conformational change within SCF^Met30 that triggers autoubiquitylation. Schematic of cross-linked residues in Met30 found with mass spectrometry under normal growth conditions and during cadmium stress. The width of lines indicates the quantity of peptides shown in Supplementary Fig. 3I. **D** The fourth ligand for cadmium binding outside of the F-box motif. Met4 deubiquitylation in C228S mutants is blocked. Strains were cultured in YEPD media, treated with 100 μM CdCl$_2$, and samples were harvested at indicated time points. Whole-cell lysates were analyzed by western blot using a Met4 antibody to follow the ubiquitylation status of Met4. **E** Dissociation kinetics of Met30 from the SCF core ligase are

decreased in C228S mutants. Cells were cultured at 30 °C in YEPD medium and treated with 100 μM CdCl$_2$ and samples were harvested at indicated time points. ^12xMycMet30 was immunoprecipitated (Met30 IP) and co-precipitated Skp1 was analyzed by western blot. Densitometric analysis of western blots is shown in Supplementary Fig. 4B. For quantifications the signal intensity for WT Met30 variant at time point 0 was set to 1 and Skp1 signals were normalized to ^12xMycMet30 to quantify Met30 dissociation from the core ligase ($n = 3$ independent experiments), data are represented as mean ± SD, $p$ values calculated by two-tailed student's $t$-test: *$p < 0.1$, **$p < 0.05$, ***$p < 0.01$. **F** Decreased cadmium-induced gene expression but unaffected methionine starvation response in the C228S mutant. Strains were grown at 30 °C in YEDP media. For cadmium treatment, cells were exposed to 100 μM CdCl$_2$ for 40 min. For methionine starvation, cultures were washed with water and shifted to minimal medium without methionine for 30 min and then harvested. RNA was extracted and expression of Met4 target genes *MET25* and *GSH1* was analyzed by RT-qPCR and normalized to 18S rRNA levels ($n = 3$ independent experiments), data are represented as mean ± SD, $p$ values calculated by two-tailed student's $t$-test: *$p < 0.1$, **$p < 0.05$, ***$p < 0.01$. Results shown in **A, B, D, E** are representative blots from three independent experiments. Source data are provided as source data files for (**E, F**).

Met30ΔF-box is very unstable in cells, but despite the low steady-state levels it was obvious that cadmium binding was abolished when the F-box was deleted (Fig. 5D and Supplementary Fig. 5B). We next changed all three conserved cysteine residues located in the F-box domain (201, 205 and 211) (Fig. 2A) to serine ("CS" in Fig. 4A and Supplementary Fig. 3A) and determined cadmium binding. The Met30^CS mutant exhibited similar expression levels to wildtype Met30 but lost the ability to bind cadmium (Fig. 5D). Cysteines 201, 205, and 211 all contributed to cadmium binding because mutation of each individual F-box cysteine reduced binding to Cd-NTA (Supplementary Fig. 5C). Together these results strengthen out hypothesis that the F-box motif is the element in Met30 that mediates cadmium binding.

To further characterize cadmium binding capabilities of Met30 we performed a competition experiment by preincubating whole cell lysates with increasing amounts of CdCl$_2$ before performing the Cd-NTA binding assay (Supplementary Fig. 5D). The titration of cadmium revealed decreased binding of Met30 with increasing cadmium concentration (Supplementary Fig. 5E). When we performed this competition Cd-NTA binding assay with lysates prepared from cells expressing the Met30^CS mutant, no change of binding was observed, confirming specific requirement of these conserved cysteine residues in the F-box domain for cadmium binding (Fig. 5E). These competition experiments also excluded precipitation of Met30 upon addition of high cadmium concentrations as an experimental artifact that could be misinterpreted as binding to Cd-NTA.

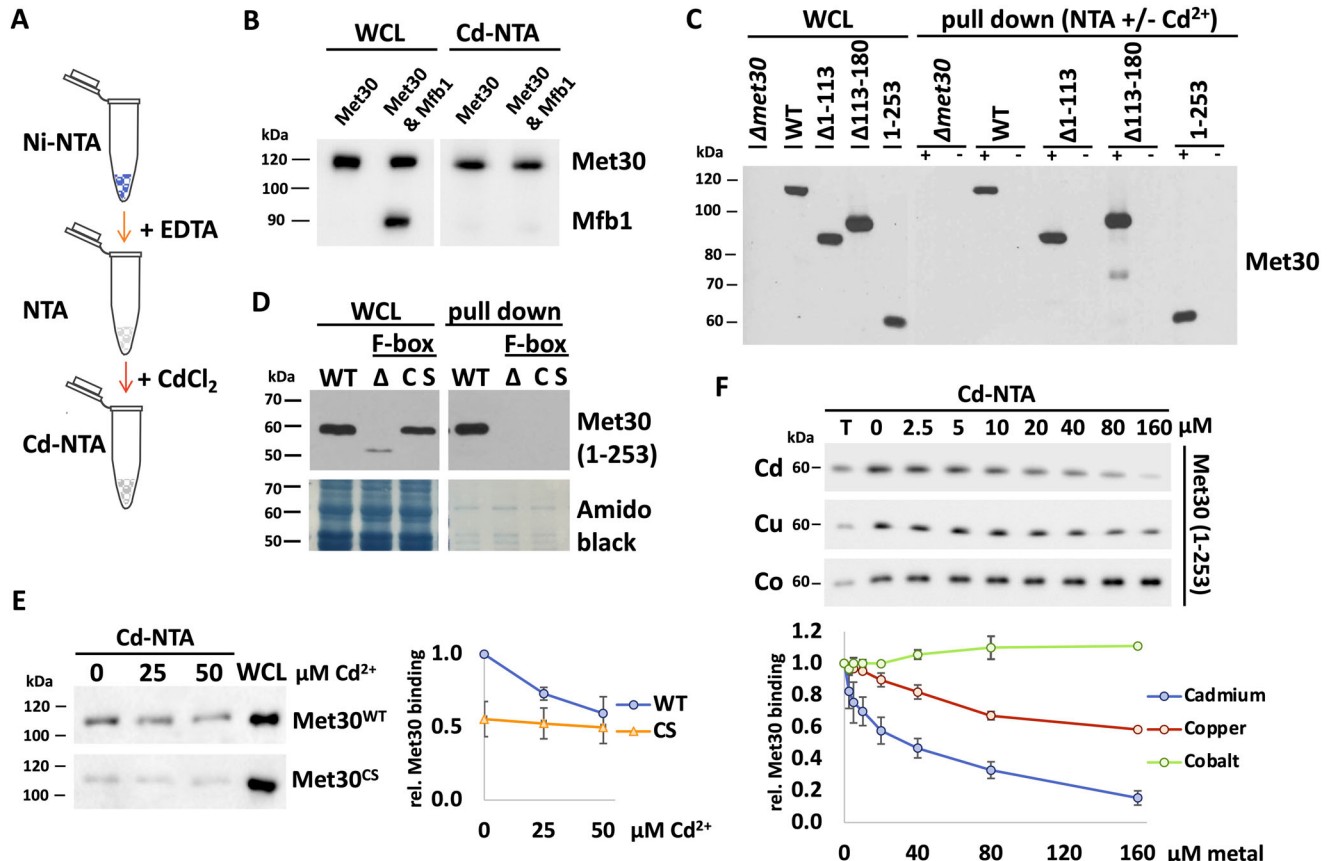

**Fig. 5 | The F-box domain of Met30 binds cadmium. A** Schematic for generation of Cd-NTA. Ni-NTA beads were stripped with EDTA and thoroughly washed with NaCl and water. Stripped NTA was incubated with CdCl₂ and beads were washed with water. **B** Binding of Met30 to Cd-NTA is specific. Cells expressing endogenous [12xMyc]Met30 exclusively or in combination with [12xMyc]Mfb1 were grown in YEPD medium, a denaturing whole cell lysate was prepared and incubated with cadmium-NTA or stripped NTA respectively overnight. An equivalent of a 10x pull down was loaded and samples were analyzed by western blot. Amido black stain is shown as a loading control. **C** Met30 truncation and internal deletion mutants do not show altered binding affinity towards cadmium-NTA compared to WT Met30. Strains introduced in Fig. 1D were cultured at 30 °C and binding to Cd-NTA was analyzed as above. **D** The F-box motif in Met30 mediates binding to cadmium in vitro. Denaturing whole cell lysates (WCL) of cells expressing C-terminally truncated Met30 (aa 1–253) with a WT F-box motif, F-box deletion or mutation of C201S, C205S & C211S (CS) in the F-box motif were prepared and incubated with Cd-NTA overnight. An

equivalent of a 10x pull down was loaded and the amido black stain of the membrane is shown for equal loading. **E** No further competition of binding with cadmium chloride without key cysteine residues in the F-box motif. Native whole cells lysates of cells expressing endogenous [12xMyc]Met30 with a WT F-box motif or mutation of C201S, C205S & C211S (CS) were prepared and incubated with indicated amounts of cadmium for 30 min. Extracts were incubated with Cd-NTA under denaturing conditions for 90 min. An equivalent of a 5x pull down (P) was loaded (n = 3 independent experiments), data are represented as mean ± SD. **F** Met30 exhibits the highest affinity towards cadmium. Experiments as in (**E**), but pre-incubation with cadmium, copper or cobalt chloride for 30 min followed by binding to cadmium-NTA for 90 min. An equivalent of a 40x pull down (P) was loaded. Densitometric analysis of western blot band intensities of Cd-NTA pull downs to respective total signals (n = 3 independent experiments), data are represented as mean ± SD. Results shown in B-F are representative blots from three independent experiments. Source data are provided as source data files for (**E**, **F**).

Disassembly of SCF[Met30] is only triggered by cadmium, but no other metals[17]. To ask if this specificity is reflected by selective Met30 metal binding, we performed the in vitro competition assay in the presence of metals other than cadmium. Whole-cell lysates of cells expressing Met30[1-253] were pretreated with cadmium, cobalt, or copper. Cobalt had no, and copper had a very modest effect on Met30 cadmium binding (Fig. 5F). However, pretreatment with cadmium potently attenuated binding. These results suggest that the F-box motif of Met30 confers specificity towards cadmium over other metals and explains the observed cadmium selectivity of the in vivo response[17].

**The Met30 F-box directly binds cadmium**
Denaturing conditions were used to exclude indirect binding by protein complexes during the cadmium-NTA pull-down assay. However, we also wanted to address cadmium binding of Met30 and the effects of F-box motif deletion in native conditions. The previously introduced cadmium binding assay on cadmium-NTA beads focused on direct

binding of F-box proteins and tried to exclude indirect effects through denaturing of protein complexes (Fig. 5). Non-specific background binding is a general problem with metal-chelate affinity chromatography under native conditions. Nevertheless, native binding experiments with the individual four cysteine ligand mutants (C201, C205, C211, and C228S) confirmed previous results obtained under denaturing conditions. All mutations significantly reduced recognition of cadmium by intact Met30-containing protein complexes (Fig. 6A).

To analyze cadmium binding of Met30 in native conditions independent of protein complexes formed in vivo, we purified recombinant Met30 from *Escherichia coli* (Supplementary Fig. 6A–C). Experiments were done in the context of recombinant MBP-Met30[1-253] fusions with and without the F-box motif. To confirm previous results obtained with Met30 purified from yeast we first analyzed recombinant purified proteins under denaturing conditions (Fig. 6B). Met30 containing the F-box motif (WT) readily purified on Cd-NTA, whereas binding was abolished in the absence of the F-box motif. Similarly, cadmium binding of recombinant Met30 in native conditions was

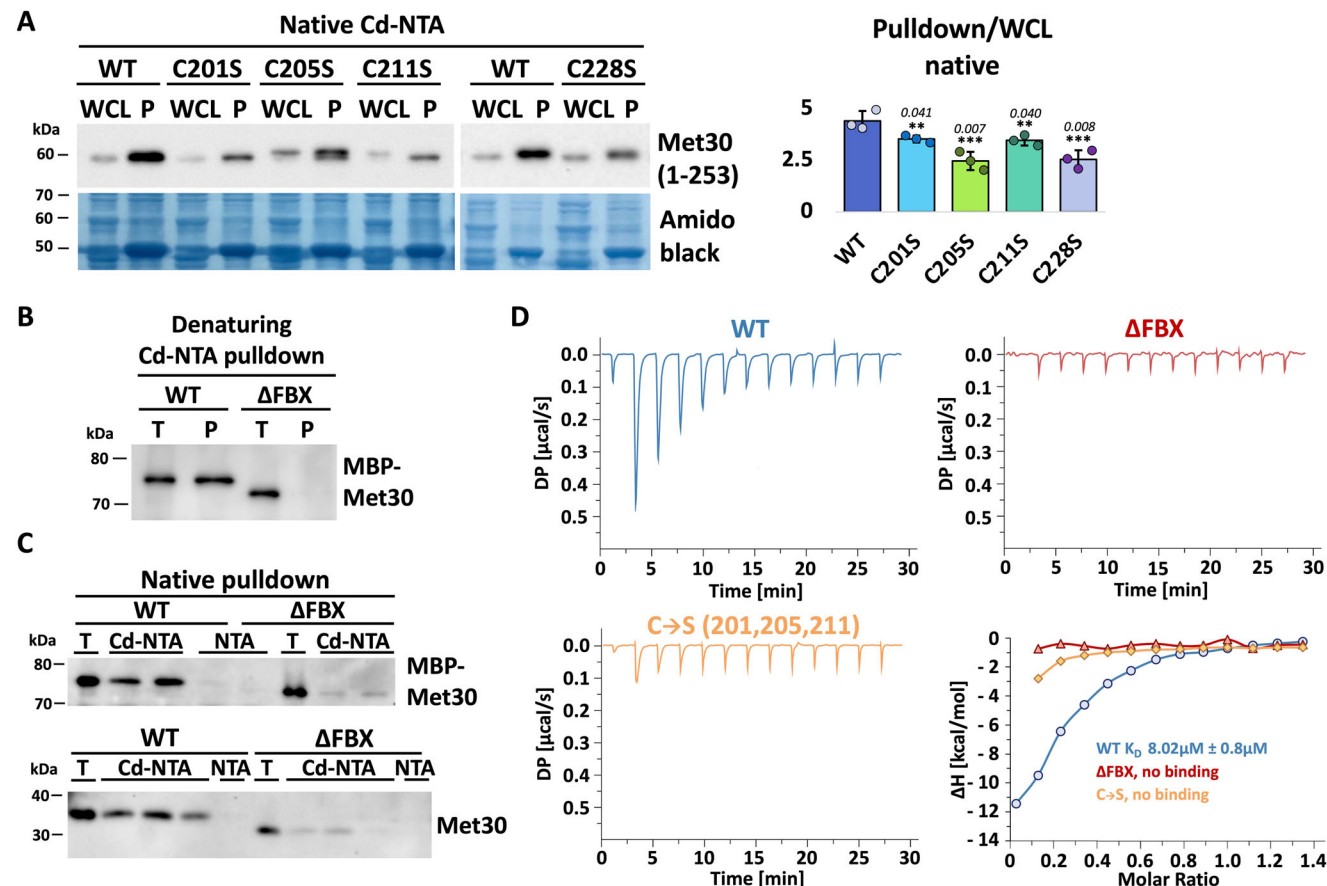

**Fig. 6 | The Met30 F-box directly binds cadmium. A** Single cysteine mutations in the F-box motif decrease cadmium binding affinity also in native conditions. Native whole cell lysates (WCL) of cells expressing C-terminally truncated Met30 (aa 1–253) with a WT F-box motif or C201S, C205S, C211S, C228S mutation respectively were prepared and incubated with cadmium-NTA for 90 min. An equivalent of a 20× pull down (P) was loaded. Densitometric analysis of Western blot band intensities of Cd-NTA pull downs normalized to respective total signals ($n = 3$ independent experiments), data are represented as mean ± SD, $p$ values calculated by two-tailed student's $t$-test: *$p < 0.1$, **$p < 0.05$, ***$p < 0.01$. **B** Purified, recombinant MBP-Met30 (T) containing a WT F-box motif or with the F-box motif deleted was incubated for 90 min with cadmium-NTA under denaturing conditions. An equivalent of

20× pull down (P) was loaded. **C** Purified, recombinant TEV digested Met30 (T) containing a WT F-box motif or with the F-box motif deleted was incubated for 90 min with cadmium-NTA or stripped NTA respectively under native conditions. Proteins were eluted in the presence of 400 mM imidazole and an equivalent of a 40x pull down was loaded. **D** ITC measurement for the titration of cadmium into recombinant MBP-Met30 variants (WT = blue, ΔFBX = red, C→S = light orange) showing a $K_D$ of 8 μM for WT Met30 and no binding for both F-Box mutants. Results shown in **D** are representative graphs from three independent experiments. Results shown in **A**–**C** are representative blots from three independent experiments. Source data are provided as source data files for (**A**).

observed with WT recombinant Met30, but binding was significantly reduced when the F-box was deleted. Identical results were obtained when the MBP fusion was removed excluding any effect of MBP (Fig. 6C). Finally, to obtain biophysical evidence of cadmium binding to recombinant, purified Met30 we performed isothermal calorimetry (ITC). ITC experiments were performed in the context of recombinant MBP-Met30[180-253] (Supplementary Fig. 6D, E). This relatively small Met30 fragment maintained ability to interact with Skp1 indicating functional folding (Supplementary Fig. 6G). The titration confirmed the interaction between the F-box and cadmium and revealed 8 μM binding affinity for wildtype Met30. In contrast, both the F-box deletion and the cysteine to serine mutant of recombinant Met30 did not bind cadmium. Together these results provide strong evidence that the F-box motif is the element in Met30 that mediates cadmium binding (Fig. 6D and Supplementary Fig. 6F).

In summary, these results demonstrate that the conserved cysteines in the F-box of Met30 are necessary for cadmium binding and are part of the cadmium sensing mechanism that initiates SCF[Met30] disassembly and activation of stress defense programs. Furthermore, these findings uncover potential functions for the F-box motif in addition to mediating protein-protein interactions with Skp1 (Fig. 7).

## Discussion

SCF ubiquitin ligases play crucial roles in many biological processes throughout the eukaryotic kingdom. From cell cycle regulation, circadian rhythms, plant hormone sensing to regulation of stress response programs, SCFs are critical for cellular and organismal homeostasis. The defining element of SCF ubiquitin ligases is a highly conserved structural motif, the F-box domain, which guides the assembly of the vast array of different SCF complexes by mediating the F-box protein-Skp1 interactions. Here we demonstrate an additional function for the F-box motif of SCF[Met30] as a sensor for environmental stress.

SCF ubiquitin ligases have been shown to monitor plant hormones[37,38], metabolites[39], and iron[40], but always through dedicated domains well separated from the F-box motif. The F-box domain itself has not been linked to environmental sensing so far, and given the high degree of sequence conservation of F-box domains our results were unexpected. When we consider the vast number of F-box proteins (23 yeast, 69 human, and over 700 Arabidopsis), it is likely that other F-box domains have sensor functions as well. The human F-box protein ßTrCP is involved in various cellular processes such as cell cycle control and is the closest mammalian homolog of Met30[20,41]. However, the

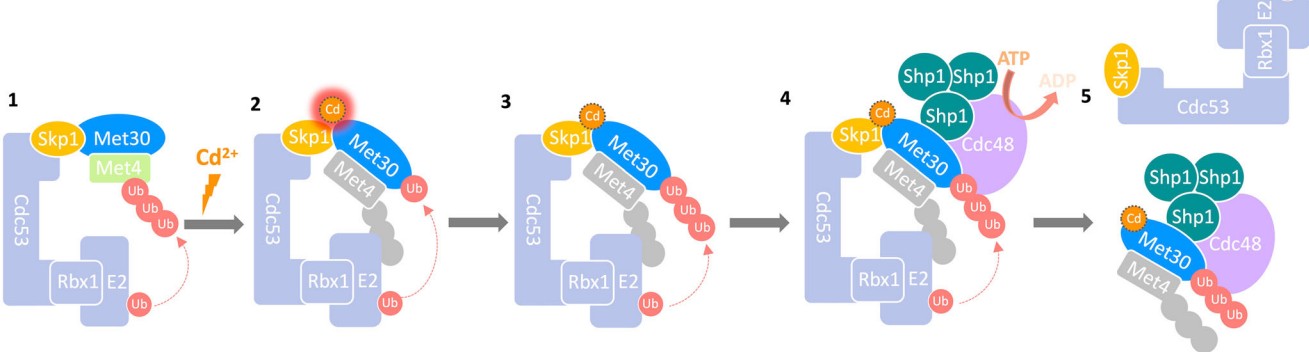

**Fig. 7 | Model for cadmium-induced SCF ligase disassembly.** (1) Under normal growth conditions, SCF mediates ubiquitylation of its substrate Met4. The transcriptional activator is kept stable and in an inactive state by the attached ubiquitin chain. (2) Cadmium binds to the F-box of Met30, which leads to a conformational change in the complex. (3) This structural change likely triggers autoubiquitylation of the F-box protein Met30. (4) The ATPase Cdc48 and its cofactor Shp1 are recruited to autoubiquitylated Met30[15]. Trimerization and therewith, local increase of Shp1 concentration on SCF[Met30] leads to Cdc48 ATPase stimulation[30]. (5) Cdc48 actively dissociates Met30 from the Cul1-Skp1 complex leading to the inactivation of the E3 ligase.

amino acid sequence of ßTrCP's F-box motif does not show similarity to the conserved, cysteine-rich cluster of Met30. Interestingly, alignment of the human F-box motives shows that several contain three or more differentially spaced cysteine residues (Supplementary Fig. 7A), which could potentially function as metal sensors. In contrast, Met30 has the only F-box domain with a cysteine cluster in yeast (Fig. 2A). Met30 shows specific binding to cadmium in vitro (Fig. 5F) and only responds to cadmium and not other stressors in vivo[17]. While we did not observe significant binding of copper or cobalt to the Met30 F-box (Fig. 5F), similar competition/pulldown assay indicated binding of zinc (Supplementary Fig. 6H). These experiments suggest, that Met30 initially has a higher affinity towards zinc, that however plateaus (Supplementary Fig. 6H) and cannot completely compete with cadmium binding. These results suggest that zinc and cadmium share some binding residues, but binding sites are not identical. We next determined the dissociation constant for Zn binding to Met30 in ITC experiments as around 1 μM. The F-box motif was required for Zn binding (Supplementary Fig. 6I). However, when yeast cells were cultured with excess amounts of zinc the Met30/Skp1 interaction was unaffected, nor could zinc preincubation affect cadmium-induced dissociation of Met30 from the core ligase (Supplementary Fig. 6J). Of note is that intracellular concentrations of free Zn (Zn available for binding) are tightly controlled and in the picomolar/nanomolar range[42], which may explain the lack of zinc-induced effects on SCF[Met30] in vivo. An alternative hypothesis is that zinc is bound to the F-box region of Met30 under normal growth conditions. Once cadmium reaches a critical concentration, zinc is replaced by cadmium, which is accompanied by a conformational change.

The three-dimensional configuration of metal complexing cysteines and contributions from other residues are likely to determine selectivity towards specific metals, and it is thus tempting to speculate that F-box motifs with different cysteine arrangements could function in selective detection of a variety of metals. Conceptually F-box embedded sensors may function in a similar mode as the cadmium sensor in Met30, not by directly disrupting the F-box/Skp1 interaction, but by inducing a conformational change that repositions the F-box protein for autoubiquitylation, which then triggers Cdc48/p97 mediated extraction of the F-box protein[15]. Direct disruption of the F-box/Skp1 interaction by environmental toxins or metabolites is also feasible, but such mechanisms are perhaps more likely to prevent formation of interactions rather than directly inducing dissociation of the complex. The Cdc48/p97 catalyzed mechanism that was demonstrated for the cadmium stress response leads to a more rapid response as it can inactivate existing complexes[15,30].

The selectivity of the SCF[Met30]/Met4 stress response for cadmium exposure is surprising but consistent with previous results[17]. Typically stress response pathways sense the stress and not the stressor, such as cadmium. For example, similar to most pathways sensing oxidative stress, cysteine oxidation of the ubiquitin ligase CRL3[Keap1] triggers response pathways to most oxidative agents including cadmium and hydrogen peroxide[43,44]. SCF[Met30] does not monitor oxidation and consequently is unresponsive to hydrogen peroxide[17]. Instead, SCF[Met30] specifically detects stress induced by cadmium, by directly binding the stressor cadmium. Nevertheless, in many ways the CRL3Keap1/Nrf2 and SCF[Met30]/Met4 pathways have overlapping purpose. Both pathways respond to nutrient and oxidative stress, but SCF[Met30]/Met4 is selective for changes in sulfur amino acid metabolism and cadmium stress with dedicated stressor sensing mechanisms, whereas CRL3Keap1/Nrf2 triggers a general response to nutrient limitation and oxidative environments. Why such a specific cadmium sensing mechanism has developed is unclear, but it may have evolved from mechanisms that monitor abundance of other metals.

SCF[Met30] forms the control hub for the response to methionine starvation and cadmium stress in yeast. Its most critical substrates are the transcriptional activator Met4 and the cell cycle inhibitor Met32, although other substrates have more recently been identified such as the centromeric histone H3 variant Cse4[45] and the autophagy protein Atg9[46]. Met4 and Met32 coordinate a cell cycle checkpoint arrest with a transcriptional response program during nutritional or cadmium stress. Surprisingly, mechanisms that activate Met4 and Met32 in response to methionine limitation or cadmium stress are distinct (Fig. 1A). Exposure to cadmium triggers the autoubiquitylation of Met30, which initiates Cdc48/Shp1 mediated disassembly of SCF[Met30] and thereby inactivation of all SCF[Met30] functions[15,30]. Methionine stress blocks Met4 binding to SCF[Met30] and thus only blocks ubiquitylation of substrates that depend on Met4 for recruitment to SCF[Met30], such as Met32, but allows ubiquitylation of other substrates to continue (Fig. 1A). The diverse mechanisms of responding to methionine and cadmium stress are also reflected in clearly separated sensor mechanism, with the F-box domain serving as a dedicated cadmium sensor and two regions containing residues 95-99 and 236-239, respectively, responding to methionine stress. Why the cadmium stress response has evolved to block ubiquitylation of all SCF[Met30] substrates is not clear. The histone H3 variant Cse4 is so far the only well-characterized substrate that is recruited to SCF[Met30] independently of Met4, and thus selectively stabilized in cadmium but not methionine stress[22,45]. However, there may be several so far uncharacterized Met4-independent substrates, and once identified they may provide a

rationale why and how the cadmium stress response is coordinated by the synchronized stabilization or activation of all these SCF^Met30 substrates.

Alterations in SCF function have been linked to neurodegenerative diseases and several types of cancer[47–49]. Hence, SCF ubiquitin ligases are very attractive drug targets, but due to their similarity, inhibition of individual ligases is challenging. This study indicates that the F-box domain may present a promising avenue to target SCF subtypes to induce their disassembly in vivo. Indeed, the first subtype selective SCF inhibitor that was identified, the small molecular enhancer of rapamycin 3 (SMER3), may work through this mode of action. SMER3 is a selective inhibitor of SCF^Met30 and induces dissociation of Met30 from the Cul1-Skp1 complex without affecting any other SCF complexes[50]. F-box protein dissociation is also observed in response to a SCF^Skp2 inhibitor[51], supporting this mode of action as a viable strategy for the development of subtype-selective SCF inhibitors.

In summary, our results demonstrate that the F-box domain has functions beyond forming the F-box protein-Skp1 interaction. Future experiments will show how common it is for F-box motifs to act as sensors for environmental or intracellular cues.

## Methods

### Plasmids, yeast strains, and growth conditions
Yeast strains used in this study are isogenic to 15DaubD, a bar1D ura3Dns, a derivative of BF264-15D[52]. Specific strains are listed in the Supplementary Information in Supplementary Table 1.

### CRISPR vectors and guide RNAs
For CRISPR/Cas9 edited yeast strains the vector pML107[53] which contains both sgRNA and Cas9 expression cassettes was used. To generate the guide RNA sequences the "CRISPR Toolset" (http://wyrickbioinfo2.smb.wsu.edu/crispr.html) was consulted. pML107 was linearized using SwaI and hybridized primers for gRNAs (see Supplementary Table 1) were inserted via Gibson assembly (GA) to generate pML107-Met30-guideRNA vectors. The vectors were transformed together with the hybridized 90mer oligos containing the Met30-specific repair sequence into PY1073, PY1314 or PY2231 respectively to generate marker-free ^12xMYCMet30 or ^HBTHMet30 mutants.

### Expression vector for recombinant Met30
PCR and GA or restriction digestion followed by ligation were used to generate pET28 MBP-TEV-Avi-Flag-Met30 1–253 (Δ231–239) vectors. To add the Avi and Flag tag, Met30 was amplified from pYLEU_met30_12xMYC_Met30 wt with primers 310 & 311. Addgene Plasmid 69929 was used to amplify the expression vector backbone with primers 312 & 313. In this process, the C-terminal 6-His tag was deleted from the original construct. GA was used to generate pET28 MBP-TEV-Avi-Flag-Met30 1-253wt out of both PCR fragments. Met30ΔFBX was amplified from pYLEU_met30_12xMYC_Met30 1–253 ΔFBX with primers 032 & 311. PCR fragment and pET28 MBP-TEV-Avi-Flag-Met30 1–253wt were digested using BamHI and NotI and ligation was used to generate pET28 MBP-TEV-Avi-Flag-Met30 1–253ΔFBX. For better solubility in the native purification, aa 231–239 of Met30 were deleted from both constructs. For wt, pET28 MBP-TEV-Avi-Flag-Met30 1–253wt was amplified using primers 314 & 315. For ΔFBX, pET28 MBP-TEV-Avi-Flag-Met30 1–253ΔFBX was amplified using primers 314 & 316. Gibson assembly was used to generate final vectors.

Generated plasmids and strains were verified by sequencing. All strains were cultured in standard media, and standard yeast genetic techniques were used unless stated otherwise[54]. References to the use of cadmium ($Cd^{2+}$) are specifically to cadmium chloride ($CdCl_2$). For cell spotting assays, strains were cultured to logarithmic growth phase, sonicated and counted. Serial dilutions were made and spotted onto YEPD plates supplemented with or without indicated amounts of $CdCl_2$ via a pin replicator (V&P Scientific, San Diego, CA).

### Protein analysis
For Western blot analyses and immunoprecipitation assays, yeast whole cell lysates were prepared under native conditions in Triton X-100 buffer (50 mM HEPES, pH 7.5, 0.2% Triton X-100, 200 mM NaCl, 10% glycerol, 1 mM dithiothreitol, 10 mM Na-pyrophosphate, 5 mM EDTA, 5 mM EGTA, 50 mM NaF, 0.1 mM orthovanadate, 1 mM phenylmethylsulfonyl fluoride [PMSF], and 1 mg/ml each leupeptin, and pepstatin). Cells were homogenized in a screw-cap tube with 0.5 mm glass beads and Antifoam Y-30 using a MP FastPrep 24 (speed 4.0, $3 \times 20$ s). Lysates were separated from glass beads and transferred into 1.5 ml reaction tubes (USA Scientific). Lysates were cleared by centrifugation (10 min, $14,000 \times g$ at 4 °C). For Western blot analyses proteins were separated by SDS-PAGE and transferred to a polyvinylidene difluoride (PVDF) membrane. Proteins were detected with the following primary antibodies: anti-Met4 (1:20,000; a gift from M. Tyers), anti-Skp1 (1:5000; a gift from R. Deshaies), anti-Cdc48 (1:20000; a gift from E. Jarosch), anti-Myc (1:2000; Santa Cruz, 9E10), anti-HA (1:2000; Santa Cruz, F7) anti-RGS6H (1:2000; QIAGEN, Germantown, MD) anti-Tubulin (1:2000; Santa Cruz). Western blots results shown are representative blots from three experiments with independent cultures. Band intensities of immunoblots were quantified using either ImageJ or the Biorad Image Lab software. For immunoprecipitation of 12xMYC tagged proteins, 1 mg of total protein lysate was incubated in Triton X-100 buffer equilibrated MYC-trap beads (Chromotek) in a final volume of 500 μl overnight at 4 °C on a nutator. The next day beads and supernatants were separated by centrifugation (2 min, $100 \times g$ at 4 °C). Beads were washed 3 times in 1 ml Triton X-100 buffer at 4 °C. Beads were resuspended in 2× Laemmli buffer and boiled for 5 min to elute proteins.

Cadmium-NTA assays: 1 ml of Nickel-NTA (Quiagen) was washed with 10 bead volumes (bv) $H_2O$ followed by a 10 min incubation in 10 bv 0.5 M NaCl at RT on a nutator. Beads were washed in 10 bv $H_2O$ and then incubated for $2 \times 30$ min in 50 mM EDTA at RT on a nutator. Beads were washed in $2 \times 10$ bv $H_2O$ and then incubated for 60 min in 100 mM $CdCl_2$ at RT on a nutator. Beads were washed twice in 10 bv $H_2O$ and resuspended in 20% EtOH and stored at 4 °C.

Denaturing protein extracts were prepared with 8 M Urea buffer (8 M Urea, 150 mM NaCl, 50 mM Tris pH 7.4, 2% SDS, 1 mM dithiothreitol, 1 mM phenylmethylsulfonyl fluoride [PMSF], and 1 mg/ml each leupeptin, and pepstatin). Cells were homogenized in a screw-cap tube with 0.5 mm glass beads and Antifoam Y-30 using a MP FastPrep 24 (speed 4.0, $3 \times 20$ s). Lysates were separated from glass beads and transferred into 1.5 ml reaction tubes (USA Scientific). Lysates were cleared by centrifugation (10 min, $14,000 \times g$ at RT). Five hundred micrograms of denatured protein extracts were incubated with 25 μl of Cd-NTA slurry. Depending on experimental set up, lysates were incubated for 90 min or overnight on a nutator at room temperature. Beads and supernatants were separated by centrifugation (2 min, $100 \times g$ at RT). Beads were washed 3 times in 1 ml 8 M Urea buffer at RT. Beads were resuspended in 2× Laemmli buffer and boiled for 5 min to elute proteins.

For metal competition assays, native whole-cell lysates were prepared as described above. Five hundred micrograms lysate was incubated with the according concentration of metal for 30 min at 4 °C on a nutator. Denaturing buffer was added to a final concentration of 6.4 M Urea and reactions were incubated for 90 min on a nutator at room temperature. Beads were washed 3 times in 1 ml 8 M Urea buffer at RT. Beads were resuspended in 2× Laemmli buffer and boiled for 5 min to elute proteins.

Native Cd-NTA pulldown with recombinant Met30: Each plasmid (pET28-MBP-TEV-Avi-Flag-Met30 1–253 (Δ231–239) wt or ΔFBX; see above) for protein expression in *E. coli* was transformed into Rosetta-BL21 (DE3) cells. Protein expression was induced with 0.15 mM IPTG for 16 h at 16 °C. Cells were harvested at 5000 rpm for 10 min and

disrupted by sonication on ice in Triton X-100 buffer (50 mM HEPES, pH 7.5, 0.2% Triton X-100, 200 mM NaCl, 10% glycerol, 1 mM dithiothreitol, 0.1 mM orthovanadate, 1 mM phenylmethylsulfonyl floride [PMSF], and 1 mg/ml each leupeptin, and pepstatin). Lysates were cleared by centrifugation (20 min, 14,000 × $g$ at 4 °C) and supernatants were incubated with amylose resin overnight at 4 °C on a nutator. The next day beads and supernatants were separated by centrifugation (2 min, 100 × $g$ at 4 °C). Beads were washed 3 times in Triton X-100 buffer at 4 °C. Elution of recombinant proteins was performed in Triton X-100 buffer containing 20 mM Maltose for 20 min at RT. Purified proteins were digested with TEV for 1 h at 30 °C. Digested proteins were cleared by centrifugation (20 min, 14,000 × $g$ at 4 °C) and supernatants were diluted in Ripa buffer (150 mM NaCl, 25 mM Tris-HCl [pH 7.4], 1% NP-40, 0.5 % DOC, 0.1% SDS and 1 mM dithiothreitol) and incubated with NTA or Cd-NTA slurry respectively for 90 min at 4 °C. Beads were washed twice with Ripa buffer for 5 min at RT on a nutator. Elution was performed in Ripa buffer containing 400 mM imidazole for 20 min at RT and 1000 rpm in a thermo-shaker. Eluted proteins were cleared by centrifugation for 20 min at 14,000 × $g$ and supernatants were analyzed by Western blot and proteins were detected with HRP-Streptavidin antibodies.

## Isothermal titration calorimetry
Each plasmid (pET28-MBP-TEV-Avi-Flag-Met30 180–253 (Δ231–239) wt, C→S (201,205,211) or ΔFBX as well as pET28-MBP-TEV-Avi-Flag for protein expression in *E. coli*) was transformed into Rosetta-BL21 (DE3) cells. Protein expression was induced with 0.15 mM IPTG for 16 h at 16 °C in LB media. Cells were harvested at 5000 rpm for 10 min and disrupted by sonication on ice in Triton X-100 buffer (50 mM HEPES, pH 7.5, 0.2% Triton X-100, 200 mM NaCl, 10% glycerol, 1 mM T-CEP, 1 mM phenylmethylsulfonyl floride [PMSF], and 1 mg/ml each leupeptin, and pepstatin). Lysates were cleared by centrifugation (30 min, 16,000 × $g$ at 4 °C) and supernatants were incubated with amylose resin overnight at 4 °C on a nutator. The next day beads and supernatants were separated by centrifugation (2 min, 100 g at 4 °C). Beads were washed 3 times in Triton X-100 buffer at 4 °C. Elution of recombinant proteins was performed in 3 steps. First two elutions were each performed in 1 bv of Triton X-100 buffer (+1mM T-CEP) containing 20 mM Maltose for 20 min at RT. The third elution was performed in 1 bv Ripa buffer (25 mM Tris pH 7.4, 150 mM NaCl, 1% NP-40, 0.5% DOC, 0.1% SDS, and 1mM T-CEP) containing 40 mM Maltose for 30 min at RT. Proteins were dialyzed against 50 mM HEPES buffer, pH 7.4, containing 200 mM NaCl and 1mM T-CEP. After dialysis, proteins were diluted to a final concentration of 50 μM. CdCl$_2$ was prepared at 500 μM in the same dialysis buffer as the corresponding proteins. ITC runs were performed at 25 C on a Malvern Micro-Cal PEAQ-ITC system starting with a single 0.4 μL injection of the cadmium solution into the chamber containing the protein solution, followed by twelve 2 μl injections, with 130 s spacing between each injection and continuous stirring at 750 rpm. To determine the baseline for each experiment, a titration of the cadmium solution into buffer was performed and subtracted accordingly. The obtained raw data was analyzed using software provided by the manufacturer to determine the apparent $K_D$.

## Native $^{HBTH}$Met30 Purification for DSSO X-Linked MS
$^{HBTH}$Met30 cells were cultured in YEP + 2% galactose + 4 μM Biotin at 30 °C and treated with 100 μM CdCl$_2$ for 30 min. Native whole cells lysates were prepared in lysis buffer (50 mM Hepes pH7.5, 200 mM NaCl, 10% glycerol, 40 mM imidazole, 0.2% TritonX, 1 mM dithiothreitol, 0.1 mM orthovanadate, 1 mM phenylmethylsulfonyl floride [PMSF], and 1 mg/ml each leupeptin and pepstatin) and bound to Ni-resin. Proteins were eluted in the presence of 250 mM imidazole. For cross-linking analysis, eluted $^{HBTH}$Met30 was first bound to Streptavidin beads and then on-bead cross-linked with 0.5 mM DSSO in PBS buffer for 1 h at 37 °C. Bead-bound proteins were reduced with TCEP and alkylated with iodocetamide and digested by trypsin prior to LC MS$^n$ analysis.

LC MS$^n$ analysis of cross-linked peptides were carried out using an UltiMate 3000 RSLC coupled to an Orbitrap Fusion Lumos mass spectrometer. Samples were loaded onto a 50 cm × 75 μm Acclaim PepMap C18 column and separated over a 102 min gradient of 4% to 25% acetonitrile at a flow rate of 300 nL/min. Ions were detected by Orbitrap in MS1 from *m/z* 375–1500 using a resolution of 120 K, AGC target of 4e5, and a maximum injection time of 50 ms. Precursor ions with charge of 4–8+ in the MS$^1$ scan were selected for MS$^2$ analysis and subjected to CID with NCE 23. The fragmentation of these ions was measured in the Orbitrap at 30 K resolution with AGC target of 5e4 and maximum injection time of 100 ms. Fragment ion pairs in each MS$^2$ scan with mass difference corresponding to the difference of DSSO fragmentation moieties (31.9721 Da) were further selected and fragmented by CID with a normalized collision energy of 35% in MS$^3$. The fragmentation ions of the precursors were detected by ion trap in Rapid mode with AGC target 2e4 for a maximum injection time of 150 ms.

## Database searching and cross-linked peptide identification
MS$^3$ spectra were extracted from RAW files using MSConvert (ProteoWizard 3.0.10738) and subjected to database searching using Batch-Tag within ProteinProspector (v.6.3.3) against a *Saccharomyces cerevisiae* database consisting of 20,240 entries concatenated with an equal number of decoy sequences. The mass tolerances were set as ±20 ppm for parent ions and 0.6 Da for fragment ions. Trypsin was set as the enzyme with three maximum missed cleavages allowed. Cysteine carbamidomethylation was selected as fixed modification, while a maximum of three variable modifications were also allowed, including methionine oxidation, N-terminal acetylation, and N-terminal conversion of glutamine to pyroglutamic acid. Three defined DSSO cross-linked modification on uncleaved lysines: alkene (C$_3$H$_2$O, +54 Da), thiol (C$_3$H$_2$SO, +86 Da) and sulfenic acid (C$_3$H$_4$O$_2$S, +104 Da) were also selected as variable modifications. Search results were integrated via in-house software xl-Tools to identify cross-linked peptide pairs.

RNA and RT-qPCR: Yeast RNA was prepared using the RNeasy Plus Mini Kit (Quiagen) according to the manufacturers protocol. cDNA synthesis was performed using Super Sript II Reverse Transcriptase Kit (Invitrogen). 1.5 μg of RNA was transcribed into cDNA according to the manufacturers protocol. The synthesized cDNA was diluted 1:15 in nuclease-free water. qPCR was performed with a Biorad CFX Connect RT-PCR machine using the Biorad iTaq Universal SYBR Green SuperMix and 4 μl of the cDNA 1:15 dilution was used in a 20 μl reaction as a template. Primers were used at a final concentration of 0.2 μM. Sequences of primers are: *MET25F:* GCCACCACTTCTTATGTTTTCG, *MET25R:* AGCAGCAGCACCACCTTC, *GSH1F:* TGACAGCATCCATCAGG ACCAG, *GSH1R:* GGAAGCCAGTTTCGCCTCTTTG, 18SrRNAF: GTGGTG CTAGCATTTGCTGGTTAT, 18SrRNAR: CGCTTACTAGGAATTCCTCG TTGAA. 18S rRNA was used as a normalization control for each sample. The ΔΔCt method was used for analysis. Data are represented as mean ± SD.

## Reporting summary
Further information on research design is available in the Nature Portfolio Reporting Summary linked to this article.

# Data availability
All data discussed are included within the paper, the provided supplementary information and source data files. Mass spectrometry data were deposited on ProteomeXchange under accession PXD048194. Source data are provided with this paper.

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

## Acknowledgements
We thank S. Jentsch, T. Rapoport, P. Silver, and R. Hampton for yeast strains; R. Deshaies, W. Harper, M. Tyers, T. Sommer, and E. Jarosch for antibodies. This work was supported by the National Institute of Health grant R35GM148350 to P.K. and the Hitachi-Nomura Award to L. L. NIH R35GM145249 to L.H. and NCI T32CA009054 to A.A.

## Author contributions
L.L., K.F. and P.K. designed research; L.L. performed most of the genetics and biochemical experiments with support from A.A. and G.D.; C.Y. ran, processed and analyzed MS data L.H. designed MS experiments and was consulted for MS data analysis. L.L. and P.K. analyzed the results; L.L. and P.K. wrote the manuscript and prepared the figures.

## Competing interests
The authors declare no competing interests.
