## [Peer Review File · Nature Communications]

Cadmium binding by the F-box domain induces p97-mediated SCF complex disassembly to activate stress response programsREVIEWER COMMENTS

Reviewer #1 (Remarks to the Author):

The manuscript by Lauinger et al. describes a novel regulatory pathway by which the F-box protein Met30 is able to directly bind and sense cadmium via interactions with its F-box domain leading to disassembly of the Met30- complex in a CDC48-dependent manner and activation of Met4-dependent transcription. This is a highly exciting manuscript that elucidates the molecular basis of Met30's cadmium sensing activity and establishes a new paradigm for how F-box domains can influence protein function beyond their canonical Skp1 binding activity. The presented data are of high quality and utilize a combination of yeast genetics and in vitro biochemistry to elucidate molecular features of this novel pathway using multiple orthogonal approaches. The mutational analysis is particularly convincing both in terms of the in vitro cadmium binding assays and the in vivo phenotypes related to cadmium sensitivity. Overall, the work reported here provides important new insight into Met30 regulation that will be of broad interest to researchers interested in pathways relevant to ubiquitination and cellular responses to heavy metal exposure.

Minor points to address:

1. The authors provide in vivo crosslinking evidence that Met30 undergoes a conformational change that facilitates SCF disassembly. Examining this idea further using their in vitro system with recombinant proteins would further strengthen the manuscript.
2. The authors report the K_d for Cd binding by free Met30 to be $\sim 8 \mu\text{M}$. Do the authors know whether the intracellular concentration of cadmium is likely to reach that concentration given that only a small fraction of the environmental cadmium is transported into the yeast and much of that is likely sequestered in vacuoles?
3. The author's model should comment on whether Cd binding is likely to be reversible such that a given Met30 molecule senses free Cd and reversibly assembles or disassembles from SCF depending on its metallation status or if the metallated Met30 is permanently deactivated / degraded after its disassembly.

Reviewer #2 (Remarks to the Author):

The authors seek to understand the molecular mechanisms underlying the biological impacts of the F-box motif in mediating the calcium sensing ability of Met30, via directly binding to calcium and

mediating Met20 autoubiquitination-triggered Cdc48 interaction and compromising of SCF/Met30 E3 ligase activity. Via various biochemical and genetic approaches, the authors identified the three conserved Cys residues within Met30 F-box motif as novel calcium interacting residues and further showed that mutating these Cys residues and another Cys (228) residue next to F-box motif can disrupt calcium sensing ability of Met30 to cause compromised cellular response to calcium stress. Further studies allowed the authors to show that calcium stress leads to Met30 autoubiquitination that can recruit Cdc48, leading to dissociated Met30 from Skp1 to shut down SCF/Met30 E3 ligase activity.

The manuscript is clearly written, and the authors have utilized various approaches to gather strong experimental evidence. However, additional in-depth investigation should be carried out to validate the detailed mechanisms. In addition, the following concerns should be addressed.

- 1). Figure 2A: it will be nice for the authors to comment if Met30 mammalian homologue (such as beta-TRCP) also have calcium sensing ability?
- 2). Figure 2C: it will be nice for the authors to examine if swap Met30-Fbox motif to Grr1 or Cdc4 confer gain-of-function in calcium sensing?
- 3). Figure 3A: the authors should comment if the critical Cys residues in Met30 F-box motif can be subjected to regulation by cellular ROS levels to impact its interaction with Calcium?
- 4). Figure 4A: C211S mutant seems to have comparable ability as WT to recruit Cdc48 upon Calcium stress.
- 5). Figure 4B: it will be nice to explore or speculate if mutating the key autoubiquitination Lys residues in Met30 can block calcium stress induced Cdc48 recruitment?
- 6). Figure 4F: does docking modeling support the notion that the four critical Cys residues are in spatial proximity to associate with Calcium?
- 7). Figure 5D: will mutating C228S also reduce calcium binding?

Reviewer #3 (Remarks to the Author):

Lauinger et al review

Nat Comms

Summary- This exciting manuscript from Lauinger examines a fundamental mechanism governing regulated protein degradation by the ubiquitin proteasome system (UPS). The multi-subunit SCF family of E3 ubiquitin ligases specify protein for degradation using a so-called F-box protein, which binds substrates and recruits them to the SCF complex for ubiquitination. Conventionally, F-box proteins are

largely treated as the same, insofar as they all bind to the SCF component SKP1, tethering the F-box protein and its bound substrate to the SCF. Rather, the point of regulation for substrate ubiquitination is largely thought to occur at the interface to F-box-substrate binding. Further, the ability of F-boxes to bind the SCF complex is thought to largely depend on substrate availability. This exciting manuscript suggests that there is much additional regulation at play that remains to be discovered for the enormous family of F-box proteins.

In yeast, the F-box protein Met30 binds and ubiquitinates Met4 and Met32, and responds to a variety of environmental and stress signals, including the loss of methionine and presence of cadmium. However, the mechanisms underlying the response remain unclear. The authors show here, using a combination of biochemical and genetic evidence, that the F-box domain is itself a cadmium sensor, capable of binding cadmium and thereby releasing Met30 from the SCF, preventing the ubiquitination of its substrates. This represents a largely unexplored area of SCF family regulation. The genetic evidence in the study is absolute rock solid. There are some pieces of biochemical evidence that could be improved, after which, I am highly supportive of this study's publication. It puts forward a new way of thinking about SCF regulation that is counter to most conventional thinking, and will clearly move the field forward in this respect.

1- In Fig 3A the authors say that in response to Cad, the individual mutation of cysteine residues in the Met30 F-box domain maintain Met4 in a ubiquitinated state. However, the C201 and C205 site mutants clearly decrease ubiquitination, whereas the C211 mutant has decreased ubiquitination to start with. Similarly, in Fig 3B, they argue that each of the mutants blocks Met30 dissociation from Skp1. However, C201S and C211S both appear to dissociate from SKP1, and C211S in fact, looks very much like the WT version of Met30. Furthermore, Met30 degradation in the WCL in response to Cad exposure appears similar in the C201S and C211S mutants. This is despite the fact that both mutants have an enormous effect on Cad sensitivity (discussed below). This complicates the interpretation. The authors need to find a way to address these biochemical inconsistencies to best interpret the differences they are seeing. Perhaps the C211S mutant, based on 3B, has a more minor role, although that would be inconsistent with the various phenotypic data.

2- In Fig 3A, the authors examine Met4 ubiquitination in cells harboring the aforementioned cysteine mutations after treatment with Cad. They conclude that the yeast cells harboring cysteine mutants maintain Met4 ubiquitination, however is decreased in C201S and C205S. While it is unchanged in the C211S mutant, it also starts much lower. Is there an obvious explanation for this? I would suggest they soften their interpretation since 2 of the 3 mutants do in fact reduce ubiquitination.

3- The data in S3C, which support a role for these single amino acid changes in Met30 in the Cad response, are simply awesome. They provide incontrovertible genetic evidence that these described cysteines in the F-box domain respond to the presence of cadmium, and strongly support the notion that these amino acids are in fact the key mediators of the cadmium response. Moreover, in conjunction with the prior data, including that in Fig 3C, they support the idea that not all F-boxes are created equal and that different ones might evolve in parallel with substrate binding motifs to control the destruction of specialized substrate repertoires under physiologic conditions. I would urge the authors to at least consider moving the data in S3C to the main figure.

4- Figure 4A would benefit from having a loading control for Cdc48 and Met30.

5- The experiment in Fig 4B is difficult to assess because of the exposure of the blot, particularly evident in the smudge in the top right. I would suggest repeating this experiment to make the differences in ubiquitination clearer between the WT and mutants. Blotting for Met32 or Met4 in the lysates, as controls, would also be beneficial.

6- Can the data in Fig 4C be quantified? This represents a key piece of evidence supporting a conformational change but is very difficult to interpret as is. Is there an overall decrease in cross-linked Met30, as it appears in the graphic? Also, it would be helpful if the authors could clarify how ubiquitination and degradation might impact the interpretation of these data.

7- In Figure 5C it would be helpful to have the F-box mutants, or cysteine mutants, shown in parallel.

Minor

8- The authors mention that the two cysteine containing domains they considered are conserved. They should show this in the supplement. Also, they say these are good candidates for cadmium sensors, but it isn't clear if their reasoning is that these are conserved, or has to do with some other feature that is not discussed.

9- The authors should consider citing Choudhury et al and Paul et al, who showed that AKT and SRC can regulate the F box proteins cyclin F and bTRCP by altering their binding to SKP1, respectively. Both lend weight to the suggestion being made here.

Reviewer #4 (Remarks to the Author):

Lauinger and colleagues investigated the interaction between Met30 and Cd²⁺ ions and its role in protecting cells from exposure to this toxic metal. Specifically, they identified four cysteine residues (three in the F-box domain and the fourth proximal to it) as the ligands responsible for Cd²⁺ chelation.

By employing cross-linking mass spectrometry on HBTH-tagged Met30 in vitro, in the presence and absence of Cd²⁺, they observed a conformational change that could be linked to Met30 autoubiquitylation, leading to SCF_{Met30} disassembly.

The manuscript is well written, and the results could be convincing and of broad interest. However, I suggest the authors address the following two main points before considering the manuscript for publication.

1) The authors claim specificity in binding Cd²⁺ ions, comparing the binding capabilities of Met30 towards Cd²⁺, Co²⁺, and Cu²⁺. While I am not an expert in this field, I was surprised by the choice of

alternative metals. Why choose Co^{2+} and Cu^{2+} instead of the more common Zn^{2+} , which is in the same group as Cd, or AsNaO₂? When discussing this point, the authors referenced a paper by Kaiser's group stating, 'we show that the yeast ubiquitin ligase SCF_{Met30} plays a central role in the response to two of the most toxic environmental heavy metal contaminants, namely, cadmium and arsenic.'

2) XLMS data are crucial for supporting the conclusions presented in this manuscript. However, I was unable to review this part as I could not find the data deposited on ProteomeXchange. Is the accession number correct? Additionally, the methods do not specify the number of biological/technical replicates performed or how the data were combined to generate the final list of XLs. Differences in the list of XLs should be supported by reproducibility, especially in the DDA acquisition mode. In the table in Figure S3, there is a list of cross-links in the two conditions with some 'counts'. Are these counts the number of spectra accounting for the unique XL sites? This table also shows XLs indicating the presence of Met30 dimers in solution. Is this expected under physiological conditions? This section needs better explanation and discussion in the text.

3) In the XL method section, line 484, 'X' and ',' are missing after the word 'triton.'

4) Could the authors use the XL data to propose a model of this conformational change, possibly starting from the alpha-fold structure of Met30? The manuscript could potentially benefit from this.

Point-by point response

We would like to thank the reviewers for their insightful and productive comments. We have addressed the points raised by the reviewers as outlined below. In addition, we added experiment S6G, to address an important question that was raised during a seminar presentation of the work. Figure S6G demonstrates that the short F-box domain constructs used in ITC experiments are correctly folded and capable of interaction with Skp1. Please note that all changes/updates in the manuscript are highlighted in yellow. We thank the reviewers for their time and critically reading our manuscript. We believe the changes further improve our manuscript.

Response to Reviewer's Comments:

Reviewer #1 (Remarks to the Author)

The manuscript by Lauinger et al. describes a novel regulatory pathway by which the F-box protein Met30 is able to directly bind and sense cadmium via interactions with its F-box domain leading to disassembly of the Met30- complex in a CDC48-dependent manner and activation of Met4-dependent transcription. This is a highly exciting manuscript that elucidates the molecular basis of Met30's cadmium sensing activity and establishes a new paradigm for how F-box domains can influence protein function beyond their canonical Skp1 binding activity. The presented data are of high quality and utilize a combination of yeast genetics and in vitro biochemistry to elucidate molecular features of this novel pathway using multiple orthogonal approaches. The mutational analysis is particularly convincing both in terms of the in vitro cadmium binding assays and the in vivo phenotypes related to cadmium sensitivity. Overall, the work reported here provides important new insight into Met30 regulation that will be of broad interest to researchers interested in pathways relevant to ubiquitination and cellular responses to heavy metal exposure.

Minor points to address:

1. The authors provide in vivo crosslinking evidence that Met30 undergoes a conformational change that facilitates SCF disassembly. Examining this idea further using their in vitro system with recombinant proteins would further strengthen the manuscript.

We agree with the reviewer and are preparing to obtain detailed structural insight into conformational changes at the atomic level. We believe structural insight is beyond of the scope of the current study.

2. The authors report the K_d for Cd binding by free Met30 to be ~8 μM. Do the authors know whether the intracellular concentration of cadmium is likely to reach that concentration given that only a small fraction of the environmental cadmium is transported into the yeast and much of that is likely sequestered in vacuoles?

It is a very interesting question, but seems very challenging on the experimental end. It would be possible to measure internal levels of cadmium by using methods such as Inductively coupled plasma mass spectrometry (ICP-MS), which we have attempted but run into technical problems due to prevalence of trace amounts of cadmium in buffer solutions. Regardless, this method would not be able to distinguish between concentrations in subcellular compartments. We do believe intracellular nuclear cadmium concentrations (Met30 is a nuclear protein) reach the

relevant level at least transiently to trigger the Met30-mediated response, because we do observe a response to cadmium stress that depends on the exact residues in Met30 that are required for cadmium binding in vitro. It is unlikely that this is a coincidence.

3. The author's model should comment on whether Cd binding is likely to be reversible such that a given Met30 molecule senses free Cd and reversibly assembles or disassembles from SCF depending on its metallation status or if the metallated Met30 is permanently deactivated / degraded after its disassembly.

A pathway for regulation of F-box protein abundance (in addition to autoubiquitylation) that specifically targets F-box proteins that are dissociated from Skp1 was previously discovered by our lab (Mathur et al 2015 <https://doi.org/10.1371/journal.pgen.1005727>). In this paper we showed that F-box proteins that are dissociated from Skp1 are recognized by a specific 'Skp1-free' F-box protein degradation pathway. Hence, as Met30 dissociated from Skp1 upon cadmium exposure/binding the F-box protein is degraded via this proteasomal pathway. We have now added this information to the manuscript to give a better understanding of Met30's fate in the presence of Cadmium.

In addition, as suggested by another reviewer we also added loading controls (Whole cell lysates) for some of the IPs (Figures 4). Unfortunately, we did not have enough sample left to re-run the gels. Hence, we redid the experiments and therefore changed Figures 4 A and B and added Suppl. 3 D-H. Quantifications were adjusted accordingly.

We now also discuss how ROS can affect Cadmium sensing (Suppl. figure 4 E-G).

Reviewer #2 (Remarks to the Author)

The authors seek to understand the molecular mechanisms underlying the biological impacts of the F-box motif in mediating the calcium sensing ability of Met30, via directly binding to calcium and mediating Met20 autoubiquitination-triggered Cdc48 interaction and compromising of SCF/Met30 E3 ligase activity. Via various biochemical and genetic approaches, the authors identified the three conserved Cys residues within Met30 F-box motif as novel calcium interacting residues and further showed that mutating these Cys residues and another Cys (228) residue next to F-box motif can disrupt calcium sensing ability of Met30 to cause compromised cellular response to calcium stress. Further studies allowed the authors to show that calcium stress leads to Met30 autoubiquitination that can recruit Cdc48, leading to dissociated Met30 from Skp1 to shut down SCF/Met30 E3 ligase activity.

The manuscript is clearly written, and the authors have utilized various approaches to gather strong experimental evidence. However, additional in-depth investigation should be carried out to validate the detailed mechanisms. In addition, the following concerns should be addressed.

1). Figure 2A: it will be nice for the authors to comment if Met30 mammalian homologue (such as beta-TRCP) also have calcium sensing ability?

We now added Met30s mammalian homolog to the discussion part where we also mention other mammalian F-box proteins with highly conserved cys-rich clusters.

Just by looking at the sequence, beta-TRCP does not seem to have metal binding ability, as it only has 2 Cys residues in its F-box motif. Preliminary unbiased studies using the mammalian system do not indicate a role for beta-TRCP.

To our knowledge, the only known F-box protein with described metal binding capability is FBXL5 which is required for the maintenance of iron homeostasis by sensing changes in iron levels in mammalian cells (Thompson et al 2012 DOI: [10.1074/jbc.M111.308684](https://doi.org/10.1074/jbc.M111.308684)) The mechanism of iron binding is however very different, as this protein contains a hemerythrin-like folded region, which interacts with the metal ion.

2). Figure 2C: it will be nice for the authors to examine if swap Met30-Fbox motif to Grr1 or Cdc4 confer gain-of-function in calcium sensing?

We indeed tried to swap F-Box motifs and replaced the Cdc4 F-box with the one of Met30. However, the Met30 F-box in the context of Cdc4 does only poorly support interaction with Skp1 (see IPs below: "Met30" = Cdc4 with Met30 F-box). F-box domains are not simply interchangeable and evolved in the context of their surrounding sequences. Therefore, we were not able to further examine if there was a gain of function in cadmium sensing in yeast. In addition, other structural features on Met30 are required for Cdc48/Shp1 binding, which are unlikely to be conserved in other F-box proteins. A Cdc4/Fbox^{Met30} hybrid is unlikely to show cadmium-mediated dissociation even if it can sense cadmium stress.

3). Figure 3A: the authors should comment if the critical Cys residues in Met30 F-box motif can be subjected to regulation by cellular ROS levels to impact its interaction with Calcium?

We thank the reviewer for this interesting question/suggestion and we added additional experiments and now also discuss how ROS can affect Cadmium sensing (Suppl. figure 4 E-G). ROS can indeed render the system less sensitive to cadmium stress.

4). Figure 4A: C211S mutant seems to have comparable ability as WT to recruit Cdc48 upon Calcium stress.

We do see variability in the individual experiments amongst the different single mutants. Therefore, we added the quantification to the Suppl materials. The standard deviation is on the higher end, however the changes between wildtype and mutants are statistically significant. Below are the 3 individual experiments that were used for the quantification.

In addition, as suggested by another reviewer we also added loading controls (Whole cell lysates) for some of the IPs (Figures 4). Unfortunately, we did not have enough sample left to re-run the gels. Hence, we redid the experiments and therefore redid experiments in Figures 4 A and B and added Suppl. 3 D-H. Quantifications were adjusted accordingly.

5). Figure 4B: it will be nice to explore or speculate if mutating the key autoubiquitination Lys residues in Met30 can block calcium stress induced Cdc48 recruitment?

This is very important point and was actually analyzed in a previous paper from our lab (Yen et al 2012 <https://doi.org/10.1016/j.molcel.2012.08.015>).

We identified 6 ubiquitin acceptor lysines in response to cadmium stress. Mutation of all six identified ubiquitin acceptor lysines to arginine neither prevented cadmium-induced Met30 ubiquitylation nor was Cdc48 recruitment or Met30/Skp1 dissociation affected suggesting that these lysine residues are preferred, but not essential ubiquitylation sites and other lysine residues can compensate for loss of preferred acceptor lysines. This flexibility in acceptor lysine is frequently observed for ubiquitin modifications.

6). Figure 4F: does docking modeling support the notion that the four critical Cys residues are in spatial proximity to associate with Calcium?

The actual structure of Met30 has not been solved yet and with the currently used predictions by tools such as AlphaFold a cadmium binding site is not obvious. Assuming AlphaFold accurately predicts the Met30 conformation, this is one of the main reasons why we think a major

conformation change has to occur to form the F-box/Cd complex. We are therefore very interested in solving structures +/- cadmium. However, we hope the reviewer agrees that such experiments are beyond the scope of this manuscript.

7). Figure 5D: will mutating C228S also reduce calcium binding?
Yes, we do show reduced binding of Met30C228S in Figure 6A.

Reviewer #3 (Remarks to the Author)

Lauinger et al review
Nat Comms

Summary- This exciting manuscript from Lauinger examines a fundamental mechanism governing regulated protein degradation by the ubiquitin proteasome system (UPS). The multi-subunit SCF family of E3 ubiquitin ligases specify protein for degradation using a so-called F-box protein, which binds substrates and recruits them to the SCF complex for ubiquitination. Conventionally, F-box proteins are largely treated as the same, insofar as they all bind to the SCF component SKP1, tethering the F-box protein and its bound substrate to the SCF. Rather, the point of regulation for substrate ubiquitination is largely thought to occur at the interface to F-box-substrate binding. Further, the ability of F-boxes to bind the SCF complex is thought to largely depend on substrate availability. This exciting manuscript suggests that there is much additional regulation at play that remains to be discovered for the enormous family of F-box proteins.

In yeast, the F-box protein Met30 binds and ubiquitinates Met4 and Met32, and responds to a variety of environmental and stress signals, including the loss of methionine and presence of cadmium. However, the mechanisms underlying the response remain unclear. The authors show here, using a combination of biochemical and genetic evidence, that the F-box domain is itself a cadmium sensor, capable of binding cadmium and thereby releasing Met30 from the SCF, preventing the ubiquitination of its substrates. This represents a largely unexplored area of SCF family regulation. The genetic evidence in the study is absolute rock solid. There are some pieces of biochemical evidence that could be improved, after which, I am highly supportive of this studies publication. It puts forward a new one of thinking about SCF regulation that is counter to most conventional thinking, and will clearly move the field forward in this respect.

1- In Fig 3A the authors say that in response to Cad, the individual mutation of cysteine residues in the Met30 F-box domain maintain Met4 in a ubiquitinated state. However, the C201 and C205 site mutants clearly decrease ubiquitination, whereas the C211 mutant has decreased ubiquitination to start with. Similarly, in Fig 3B, they argue that each of the mutants blocks Met30 dissociation from Skp1. However, C201S and C211S both appear to dissociate from SKP1, and C211S in fact, looks very much like the WT version of Met30. Furthermore, Met30 degradation in the WCL in response to Cad exposure appears similar in the C201S and C211S mutants. This is despite the fact that both mutants have an enormous effect on Cad sensitivity (discussed below). This complicates the interpretation. The authors need to find a way to address these biochemical inconsistencies to best interpret the differences they are seeing. Perhaps the C211S mutant, based on 3B, has a more minor roll, although that would be inconsistent with the various phenotypic data.

We agree with the reviewer and have softened our statement regarding Met4 ubiquitylation. We would like to stress that preserving Met4 in the ubiquitylated state is experimentally challenging.

Nevertheless the reviewer is correct and we consistently see some loss of Met4 ubiquitylation in some of the mutants. It is possible that histidine residues in the F-box domain may contribute to cadmium binding in the context of the cysteine mutations. We interpreted the fact that C211S mutants show reduced Met4 ubiquitylation in unstressed cells that the change to serine slightly compromises either strength of interaction with Skp1 or induces a slight change in substrate orientation to reduce substrate ubiquitylation efficiency.

2- In Fig 3A, the authors examine Met4 ubiquitination in cells harboring the aforementioned cysteine mutations after treatment with Cad. They conclude that the yeast cells harboring cysteine mutants maintain Met4 ubiquitination, however is decreased in C201S and C205S. While it is unchanged in the C211S mutant, it also start much lower. Is there an obvious explanation for this? I would suggest they soften their interpretation since 2 of the 3 mutants do in fact reduce ubiquitination.

Point 2 was answered above.

3- The data in S3C, which support a role for these single amino acid changes in Met30 in the Cad response, are simply awesome. They provide incontrovertible genetic evidence that these described cysteines in the F-box domain respond to the presence of cadmium, and strongly support the notion that these amino acids are in fact the key mediators of the cadmium response. Moreover, in conjunction with the prior data, including that in Fig 3C, they support the idea that not all F-boxes are created equal and that different ones might evolve in parallel with substrate binding motifs to control the destruction of specialized substrate repertoires under physiologic conditions. I would urge the authors to at least consider moving the data in S3C to the main figure.

As yeast geneticists we also love the spot plates, however when we presented the data to a broad (mostly non-yeast) audience it was suggested to us to show the growth phenotypes with liquid cultures as this might be easier to understand and quantified. Therefore, we decided to put the growth curves in the absence and presence of Cd in the main figure (3E) and the spot plates in the suppl part of the manuscript.

4- Figure 4A would benefit from having a loading control for Cdc48 and Met30.

We agree with this! Unfortunately, we did not have enough sample left to re-run the gels. Hence, we redid the experiments and therefore changed Figures 4 A and used the new set of experiments and added Suppl. 3 D. Quantifications shown in Suppl. figure 3E were adjusted accordingly. We now also show Skp-1 Co-IPs in this experiment.

5- The experiment in Fig 4B is difficult to assess because of the exposure of the blot, particularly evident in the smudge in the top right. I would suggest repeating this experiment to make the differences in ubiquitination clearer between the WT and mutants. Blotting for Met32 or Met4 in the lysates, as controls, would also be beneficial.

Here we also did not have enough sample to re-run the gel, so we redid the experiment and show the new blot in Figure 4B. Additionally we added loadings controls as (WCL) and the background control in Suppl figure 3 F&G. Quantifications shown in Suppl. figure 3H were adjusted accordingly.

6- Can the data in Fig 4C be quantified? This represents a key piece of evidence supporting a conformational change but is very difficult to interpret as is. Is there an overall decrease in cross-linked Met30, as it appears in the graphic? Also, it would be helpful if the authors could clarify how ubiquitination and degradation might impact the interpretation of these data.

The reviewer brings up a very good point. We now included an overview of the total peptide counts (n=3) of Met30 below the table showing the Met30 cross links in Suppl figure 3 I. The overall Met30 peptide count in the cadmium samples is higher than in the controls. Accordingly, lower x-links due to lower Met30 levels can be excluded in the Cd sample. We also attached an Excel sheet with more data of the MS analyses, if there is interest to have a closer look into it. This data also shows decreasing amounts of Skp1 indicating the dissociation process and increased peptide count for Cdc48 showing the recruitment of the ATPase in the Cd sample. We hope this helps to better interpret this data.

Note that the graphic in figure 4C was adjusted accordingly with the thickness of the lines indicating the cross-links reflecting the number of detected peptides in the 3 independent experiments.

7- In Figure 5C it would be helpful to have the F-box mutants, or cysteine mutants, shown in parallel.

We are not certain we understand what the reviewer is referring to. If it is related to having a “positive” control for loss of cadmium binding figure 5D contains such a control. Note that both panels in 5D are from the same gel and the same exposure (please see uncropped blot 5D).

Minor

8- The authors mention that the two cysteine containing domains they considered are conserved. They should show this in the supplement. Also, they say these are good candidates for cadmium sensors, but it isn't clear if their reasoning is that these are conserved, or has to do with some other feature that is not discussed.

We do include a conservation ranking of the first 250 amino acids of Met30 in Suppl. figure 1A. The scan was done with the “ConSurf server” and highly conserved residues are highlighted in magenta. As for the proposed metal binding site: This was actually suggested to us by structural biologists. They suggested that general metal binding with those motifs might be possible in the context of the Met30 homodimer.

9- The authors should consider citing Choudhury et al and Paul et al, who showed that AKT and SRC can regulate the F box proteins cyclin F and bTRCP by altering their binding to SKP1, respectively. Both lend weight to the suggestion being made here.

We now include the studies in the manuscript.

Reviewer #4 (Remarks to the Author):

Lauinger and colleagues investigated the interaction between Met30 and Cd²⁺ ions and its role in protecting cells from exposure to this toxic metal. Specifically, they identified four cysteine residues (three in the F-box domain and the fourth proximal to it) as the ligands responsible for Cd²⁺ chelation.

By employing cross-linking mass spectrometry on HBTH-tagged Met30 in vitro, in the presence and absence of Cd²⁺, they observed a conformational change that could be linked to Met30 autoubiquitylation, leading to SCFMet30 disassembly.

The manuscript is well written, and the results could be convincing and of broad interest. However, I suggest the authors address the following two main points before considering the manuscript for publication.

1) The authors claim specificity in binding Cd²⁺ ions, comparing the binding capabilities of Met30 towards Cd²⁺, Co²⁺, and Cu²⁺. While I am not an expert in this field, I was surprised by the choice of alternative metals. Why choose Co²⁺ and Cu²⁺ instead of the more common Zn²⁺, which is in the same group as Cd, or AsNaO₂? When discussing this point, the authors referenced a paper by Kaiser's group stating, 'we show that the yeast ubiquitin ligase SCFMet30 plays a central role in the response to two of the most toxic environmental heavy metal contaminants, namely, cadmium and arsenic.'

The reviewer makes a very important suggestion, which we have considered and have some results that we can include in the manuscript.

We performed competition/pulldown assay in which Met30 is either preincubated with cadmium or zinc respectively before binding to Cd-NTA was measured. The results suggest, that Met30 initially has a higher affinity towards zinc, that however plateaus (Suppl. figure 6H) and cannot completely compete with cadmium binding. This result in our opinion suggests that Zn and Cd share some binding residues, but binding sites are not identical. We then determined the K_d of Zn binding to Met30 in ITC experiments as around 1 μM. The F-box motif was required for Zn binding (Suppl. figure 6I). However, when we grow yeast with excess amounts of zinc and then expose the cultures to cadmium, zinc preincubation did not affect dissociation of Met30 from the core ligase. (Suppl figure 6J). Of note is that intracellular concentrations of free Zn (Zn available for binding) are tightly controlled and in the picomolar/nanomolar range and might thus never reach concentrations necessary for binding to Met30 (<https://doi.org/10.1152/physrev.00035.2014>).

An alternative hypothesis is that zinc is bound/complexed in the F-Box region of Met30 under normal growth conditions. Once cadmium surpassed a certain concentration in the cell, zinc gets replaced by cadmium, which is accompanied by a conformational change.

Our goal is to test this hypothesis using a structural approach. However, we hope the reviewer agrees that such experiments are beyond the scope of this manuscript.

2) XLMS data are crucial for supporting the conclusions presented in this manuscript. However, I was unable to review this part as I could not find the data deposited on ProteomeXchange. Is the accession number correct? Additionally, the methods do not specify the number of biological/technical replicates performed or how the data were combined to generate the final list of XLs. Differences in the list of XLs should be supported by reproducibility, especially in the DDA acquisition mode. In the table in Figure S3, there is a list of cross-links in the two conditions with some 'counts'. Are these counts the number of spectra accounting for the unique XL sites? This table also shows XLs indicating the presence of Met30 dimers in solution. Is this

expected under physiological conditions? This section needs better explanation and discussion in the text.

We are sorry, that the data was not accessible. The log in should work with:

<https://www.ebi.ac.uk/pride/login>

Username: reviewer_pxd048194@ebi.ac.uk

Password: P1QyTRCu

We now included an overview of the total peptide counts of Met30 below the table showing the summary of Met30 cross links in Suppl figure 3 I (n=3, biological replicates). The overall Met30 peptide count in the cadmium samples is higher as compared to untreated samples. Accordingly, lower x-links due to lower Met30 levels can be excluded in the Cd sample. We also attached an Excel sheet with more data of the MS analysis. This data also shows decreasing amounts of Skp1 indicating the dissociation process and increased peptide count for Cdc48 showing the recruitment of the ATPase in the Cd sample. We hope this helps to better interpret this data.

Note that the graphic in figure 4C was adjusted accordingly with the thickness of the lines indicating the cross-links reflecting the number of detected peptides in the 3 independent experiments.

All sets show overall the trend of X-link changes within Met30 when Control and Cd treated samples are compared. With the trend of decreasing Intra-links in the cadmium treated samples. We have expanded the text of this section in the manuscript to better explain this experiment.

3) In the XL method section, line 484, 'X' and ',' are missing after the word 'triton.'

Thank you, we fixed this in the manuscript.

4) Could the authors use the XL data to propose a model of this conformational change, possibly starting from the alpha-fold structure of Met30? The manuscript could potentially benefit from this.

This is a good suggestion from the reviewer. However, as mentioned above we are trying to obtain high resolution structures for the Met30/metal complexes, which hopefully provide detailed insight into the structural aspects. We are not comfortable to speculate on conformational changes or configurations based on x-linked peptides at this point.

REVIEWERS' COMMENTS

Reviewer #1 (Remarks to the Author):

The authors have adequately addressed all concerns.

Reviewer #2 (Remarks to the Author):

The authors have addressed most of the raised concerns.

Reviewer #3 (Remarks to the Author):

The reviewers have addressed all of the concerns that I raised in my prior review. I am very enthusiastic about this manuscript.

Reviewer #4 (Remarks to the Author):

I commend the authors for the constructive changes made, effectively addressing my points. With these improvements, I believe the manuscript is now suitable for publication in Nature Communications.